# Cell-cell interaction determines cell fate of mesoderm-derived cell in tongue development through Hh signaling

Maiko Kawasaki[1], Katsushige Kawasaki[1,2], Finsa Tisna Sari[1], Takehisa Kudo[1,3], Jun Nihara[3], Madoka Kitamura[1,3], Takahiro Nagai[1], Vanessa Utama[1], Yoko Ishida[2], Fumiya Meguro[1], Alex Kesuma[1], Akira Fujita[3], Takayuki Nishimura[1], Yuan Kogure[1], Satoshi Maruyama[4], Jun-ichi Tanuma[4], Yoshito Kakihara[5], Takeyasu Maeda[2], Sarah Ghafoor[6], Roman H Khonsari[6], Pierre Corre[7], Paul T Sharpe[6], Martyn Cobourne[6], Brunella Franco[8,9,10], Atsushi Ohazama[1]*

[1]Division of Oral Anatomy, Faculty of Dentistry & Graduate School of Medical and Dental Sciences, Niigata University, Niigata, Japan; [2]Center for Advanced Oral Science, Faculty of Dentistry & Graduate School of Medical and Dental Sciences, Niigata University, Niigata, Japan; [3]Division of Orthodontics, Faculty of Dentistry & Graduate School of Medical and Dental Sciences, Niigata University, Niigata, Japan; [4]Division of Oral Pathology, Faculty of Dentistry & Graduate School of Medical and Dental Sciences, Niigata University, Niigata, Japan; [5]Division of Dental Pharmacology, Faculty of Dentistry & Graduate School of Medical and Dental Sciences, Niigata University, Niigata, Japan; [6]Centre for Craniofacial & Regenerative Biology, Faculty of Dentistry, Oral & Craniofacial Sciences, King's College London, Guy's Hospital, London, United Kingdom; [7]Service de Chirurgie Maxillofaciale et tomatology, Centre Hospitalier Universitaire de Nantes,1 place Alexis Ricordeau 44000, Nantes, France; [8]Telethon Institute of Genetics and Medicine (TIGEM), Pozzuoli, Naples, Italy; [9]Medical Genetics, Department of Translational Medical Sciences, Federico II University of Naples, Italy, Naples, Italy; [10]Scuola Superiore Meridionale, School for Advanced Studies, Genomics and Experimental Medicine program,, Naples, Italy

*For correspondence: atsushiohazama@dent.niigata-u.ac.jp

**Competing interest:** The authors declare that no competing interests exist.

**Abstract** Dysfunction of primary cilia leads to genetic disorder, ciliopathies, which shows various malformations in many vital organs such as brain. Multiple tongue deformities including cleft, hamartoma, and ankyloglossia are also seen in ciliopathies, which yield difficulties in fundamental functions such as mastication and vocalization. Here, we found these tongue anomalies in mice with mutation of ciliary protein. Abnormal cranial neural crest-derived cells (CNCC) failed to evoke Hh signal for differentiation of mesoderm-derived cells into myoblasts, which resulted in abnormal differentiation of mesoderm-derived cells into adipocytes. The ectopic adipose subsequently arrested tongue swelling formation. Ankyloglossia was caused by aberrant cell migration due to lack of non-canonical Wnt signaling. In addition to ciliopathies, these tongue anomalies are often observed as non-familial condition in human. We found that these tongue deformities could be reproduced in wild-type mice by simple mechanical manipulations to disturb cellular processes which were disrupted in mutant mice. Our results provide hints for possible future treatment in ciliopathies.

## Editor's evaluation

In this study, Kawasaki and colleagues made valuable finding that Hh signaling regulates mesoderm-derived cell fate determination through cell-cell interactions. They provided compelling evidence

that disruption of primary cilia can lead to tongue development defects. They also showed solid evidence that correction of ciliary function may help to treat tongue malformations.

## Introduction

The primary cilium, a nonmotile organelle existing on almost all somatic cell surfaces in vertebrates, has been shown to play critical roles in many biological processes, including regulating signaling pathways (*Evans et al., 2006*, *Bisgrove and Yost, 2006*). The primary cilium consists of a membrane-bound cylinder surrounding nine doublet microtubules, which extends from a centriole termed the basal body. Cilia are assembled and maintained by intraflagellar transport (IFT), in which multiple protein complexes are moved bidirectionally along the axoneme. Perturbations in the function of primary cilia result in a wide spectrum of human diseases: the ciliopathies (*Rosenbaum and Witman, 2002*; *Pan et al., 2005*; *Bisgrove and Yost, 2006*; *Evans et al., 2006*; *Scholey and Anderson, 2006*). Ciliopathy patients exhibit prominent mixed symptoms in several vital organs including the brain, lung, kidney and liver, and others, such as the eye and digit. Multiple congenital tongue anomalies including aglossia, cleft, hamartoma, and ankyloglossia are also observed in ciliopathy patients (*Mostafa et al., 2005*; *Auber et al., 2007*; *Zaghloul and Brugmann, 2011*). These tongue anomalies significantly impair quality of life, since the tongue plays a critical role in multiple fundamental functions including mastication, deglutition, general sensation and taste, oral cleansing, and vocalization. In addition to ciliopathies, these tongue anomalies are often observed as non-familial condition (*Cobourne et al., 2019*; *Hill et al., 2021*; *Li et al., 2020*; *Yin and Zhao, 2020*). However, the cause of these tongue anomalies remains unclear.

The tongue is a muscular organ. During tongue development, tongue myoblasts originate from the anterior-most somites (the occipital somites), which are derived from the mesoderm (*Parada et al., 2012*; *Ziermann et al., 2018*; *Sambasivan et al., 2011*). Cells from these somites (known collectively as the hypoglossal cord) passed through the posterior edge of the caudal pharyngeal arch, along the lateral side of the pericardium toward the anterior, ventral to branchial arches, and then inside the second and first branchial arches (*Mackenzie et al., 1998*; *Huang et al., 1999*; *Sambasivan et al., 2011*; *Adachi et al., 2018*). Cranial neural crest-derived cells (CNCC) also migrate to the tongue primordia, and contribute to form the lamina propria, tendon, and interstitial connective tissue to compartmentalize tongue muscles and serve as their attachments. A lingual swelling begin to appear in the mandibular processes at embryonic day (E) 11.5 in mice, which undergoes rapid enlargement, with a rudimentary tongue-like structure observed by E12.5.

OFDI syndrome is an X-linked dominant ciliopathy that affects females and causes prenatal lethality in males (*Macca and Franco, 2009*). OFD1 syndrome is characterized by malformations of the oroface and digits (*Macca and Franco, 2009*). *OFD1* has been identified as the gene mutated in OFD1 syndrome patients. The *OFD1* gene is located on the X-chromosome and the encoded protein is localized to basal bodies of primary cilia. Congenital tongue anomalies are also observed in OFD1 syndrome patients (*Mostafa et al., 2005*; *Auber et al., 2007*; *Zaghloul and Brugmann, 2011*).

Here, we describe that tongue development requires proper cell-cell contact, genetic or mechanical disruption of the cellular processes in CNCC attributes to congenital tongue malformations as familial and non-familial conditions, respectively.

## Results

### Tongue phenotypes in Ofd1 mutant

We firstly examined mice with conditional deletion of *Ofd1* in CNCC using *Wnt1-Cre*. Both hemizygous [*Ofd1^fl^;Wnt1Cre*(HM)] and heterozygous *Ofd1* mutant mice [*Ofd1^fl/WT^;Wnt1Cre*(HET)] die at birth. *Ofd1^fl^;Wnt1Cre*(HM) mice have aglossia (*Figure 1B*; *Okuhara et al., 2019*), suggesting that complete deletion of *Ofd1* in CNCC leads to a failure of tongue formation. In contrast, *Ofd1^fl/WT^;Wnt1Cre*(HET) mice have an obvious abnormal-shaped tongue, characterized by the presence of clefts and multiple protrusions (*Figure 1D*, *Figure 1—figure supplement 1*; n=58/58). These morphological anomalies varied widely in *Ofd1^fl/WT^;Wnt1Cre*(HET) mice.

Ectopic bone (confirmed by *Runx2* expression; *Figure 1—figure supplement 2B*) and sparse tissue were observed in the tongue of *Ofd1^fl/WT^;Wnt1Cre*(HET) mice, although normal muscle was

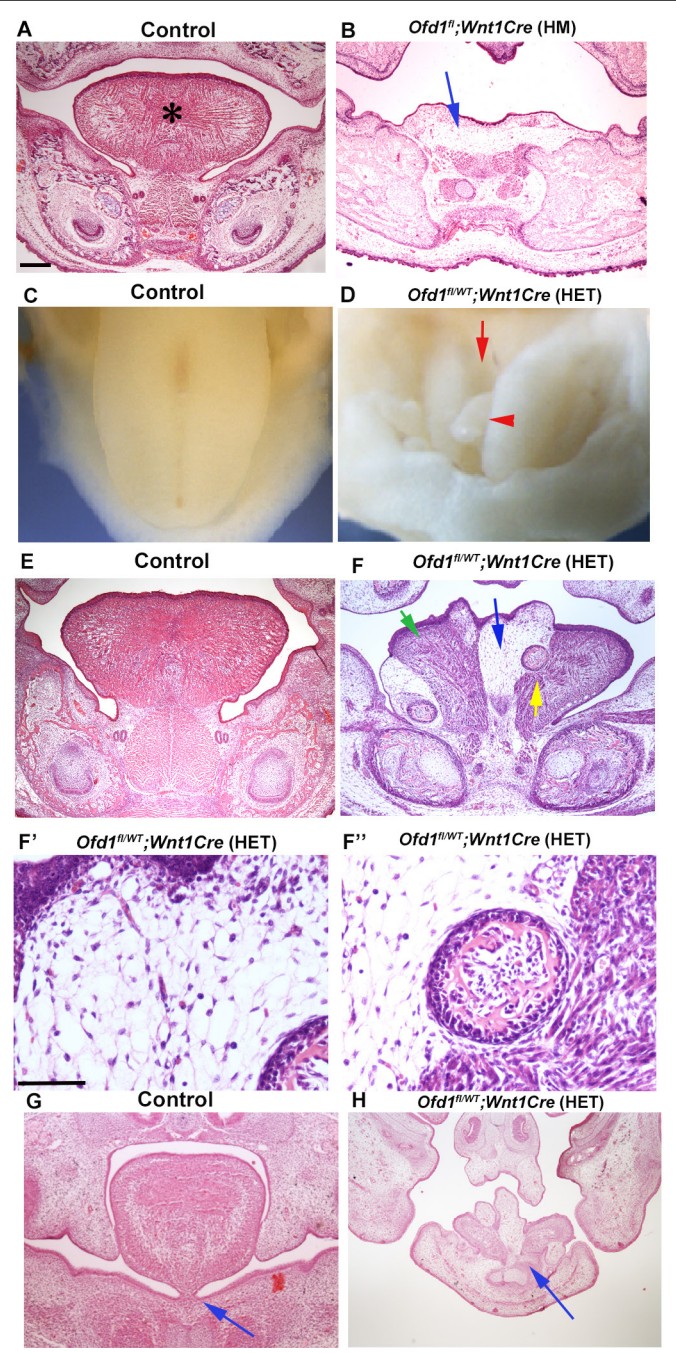

**Figure 1.** Tongue phenotypes in *Ofd1* mutant mice. (**A, B, E–H**) Frontal sections showing histological images in wild-type (**A, E, G**), *Ofd1^fl^;Wnt1Cre*(HM) (**B**) and *Ofd1^fl/WT^;Wnt1Cre*(HET) (**F–F″, H**) at embryonic day (E) 18.5. Arrow indicating sparse tissue (**B**). *: tongue (**A**). Green, blue, and yellow arrows indicating normal muscle, ectopic sparse tissue, and ectopic bone, respectively (**F**). F′ and F″ are high magnification of F indicated by blue and yellow arrow, respectively. Arrows indicating tongue frenum region (**G, H**). The presence of clefts and multiple protrusions; n=58/58. (**C, D**) Image showing oral view of tongue in wild-type (**C**) and *Ofd1^fl/WT^;Wnt1Cre*(HET) (**D**). Arrowhead and arrow indicating ectopic protrusion and cleft, respectively (**D**). Lack of tongue frenum in *Ofd1^fl/WT^;Wnt1Cre*(HET); n=30/58. Scale bars: 200 µm (A, B, E, F, G, H), 100 µm (F′, F″).

The online version of this article includes the following figure supplement(s) for figure 1:

**Figure supplement 1.** Abnormal shaped tongue in *Ofd1^fl/WT^;Wnt1Cre*(HET) mice.

**Figure supplement 2.** Bone in tongue of *Ofd1^fl/WT^;Wnt1Cre*(HET) mice.

**Figure supplement 3.** Tongue phenotypes in *Ofd1^fl/WT^;Wnt1Cre*(HET) mice.

also present (*Figure 1F–F"*). Mechanism of ectopic bone formation in *Ofd1* mutant tongue will be described elsewhere, since it is likely caused by abnormal mandibular formation, not tongue formation [*Ofd1^fl;Wnt1Cre*(HM) mice showed duplicated mandible, and bone in tongue of *Ofd1^fl/WT;Wnt-1Cre*(HET) mice was connected to mandibular bone] (*Figure 1—figure supplement 2B and D*). The size and location of ectopic sparse tissue varied widely in these mice (*Figure 1—figure supplement 3*). Protrusions consisted of sparse tissue. A lack of tongue frenum was often observed in *Ofd1^fl/WT;Wnt1Cre*(HET) mice (n=30/58; *Figure 1H*).

To identify the type of sparse tissue found in *Ofd1^fl/WT;Wnt1Cre*(HET) mice, we analyzed gene expression following laser microdissection. Results of PCR and immunohistochemistry analysis suggested that this tissue was brown adipose tissue, but not muscle or white adipose tissue (*Figure 2A–C*, *Figure 2—source data 1*, *Figure 2—figure supplements 1 and 2A*, *Figure 2—figure supplement 1—source data 1*). Molecular characters of ectopic adipose is resemble to low thermogenic brown adipose tissue, since ectopic adipose showed low expression of UCP1 and adiponectin in comparison with high thermogenic brown adipose (*Figure 2—figure supplement 2B and C*, *Figure 2—figure supplement 2—source data 1*; *Duerre and Galmozzi, 2022*). Protrusion was thus hamartoma of adipose tissue. Although mucous salivary glands associated-adipose and intermuscular adipose are known to be present in wild-type tongue, we found that the brown adipose tissue found in the *Ofd1* mutant tongue was not related to these adipose (*Figure 2—figure supplement 3*). These anomalies could not be detected in mice with epithelial conditional deletion of *Ofd1* using *Keratin (Krt) 14-Cre* mice [*Ofd1^fl/WT;Krt14Cre*(HET) or *Ofd1^fl;Krt14Cre*(HM)].

We already confirmed that *Ofd1^fl;Wnt1Cre*(HM) mice exhibit aglossia (*Figures 1B and 2E*, *Okuhara et al., 2019*). The sparse tissue was also observed in the region where the tongue usually emerged (*Figures 1B, 2E and G*). PCR analysis showed that the sparse tissue found in *Ofd1^fl;Wnt1Cre*(HM) mice was also brown adipose (*Figure 2H and I*, *Figure 2—figure supplement 4*, *Figure 2—figure supplement 4—source data 1*). Thus, abnormal brown adipose tissue formation was also occurred, when *Ofd1* is completely deleted in CNCC.

## Cell differentiation by cell-cell interaction

Brown adipose tissue is normally derived from the mesoderm (*Kajimura et al., 2010*), whereas *Ofd1* is deleted in CNCC in *Ofd1^fl/WT;Wnt1Cre*(HET) mice. In order to understand whether ectopic brown adipose tissue in the *Ofd1* mutant tongue was derived from CNCC, we examined *Ofd1^fl/WT;Wnt1Cre*(HET);R26R^YFP* mice using *ROSA26^LSL-YFP* mice. CNCC and mesoderm-derived cells were observed as YFP-positive and -negative cells, respectively, in these mice. Brown adipose marker, C/EBPβ, was not overlapped with YFP expression in *Ofd1^fl/WT;Wnt1Cre*(HET);R26R^YFP* mice (*Figure 2J*). Furthermore, brown adipose marker was overlapped with mesoderm marker molecule, Pax3 (*Figure 2L*). These indicate that ectopic brown adipose was derived from mesoderm. Tongue muscles are also derived from the mesoderm in wild-type mice, and tongue muscle in *Ofd1^fl/WT;Wnt1Cre*(HET);R26R^YFP* mice was overlapped with Pax3, but not YFP expression (*Figure 2M*). These suggest that muscular tissue in mutant tongue was derived from the mesoderm under normal conditions. Thus, ectopic brown adipose tissue was formed from mesoderm-derived cells, when *Ofd1* was deleted from CNCC. It is possible that the interaction between CNCC and mesoderm-derived cells is involved in differentiation of mesoderm-derived cells. To investigate this hypothesis, we firstly examined the early stages of tongue development to identify the timing and location of possible interactions between CNCC and mesoderm-derived cells. Morphological anomalies and obvious adipose marker expression were observed in the tongue of *Ofd1^fl/WT;Wnt1Cre*(HET) mice at E12, when the early tongue swelling has become apparent (*Figure 2O and Q*). Next, we examined *Ofd1^fl;Wnt1Cre*(HM) mice at E11. Ectopic adipose would be easily identified in these mice at such early stage, since ectopic adipose was formed as full penetrant in these mice. Myogenic progenitor cells were observed in wild-type mandibular processes, which was remarkably reduced in *Ofd1^fl;Wnt1Cre*(HM) mice at E11 (*Figure 2T and U*). Brown adipose marker was observed in *Ofd1^fl;Wnt1Cre*(HM) mice, which could not be seen in wild-type mandibular processes (*Figure 2V and W*). These indicate that mesoderm-derived cells were already demonstrating an abnormal status when they reached to mandibular processes. Thus, the timing of interactions between CNCC and mesoderm-derived cells is likely stage before E11.

At E10, mesoderm-derived cells and CNCC were observed within pharyngeal arch in the wild-type confirmed by *Mesp1Cre;R26R^YFP* and *Wnt1Cre;R26R^YFP* mice, respectively (*Figure 3A and B*).

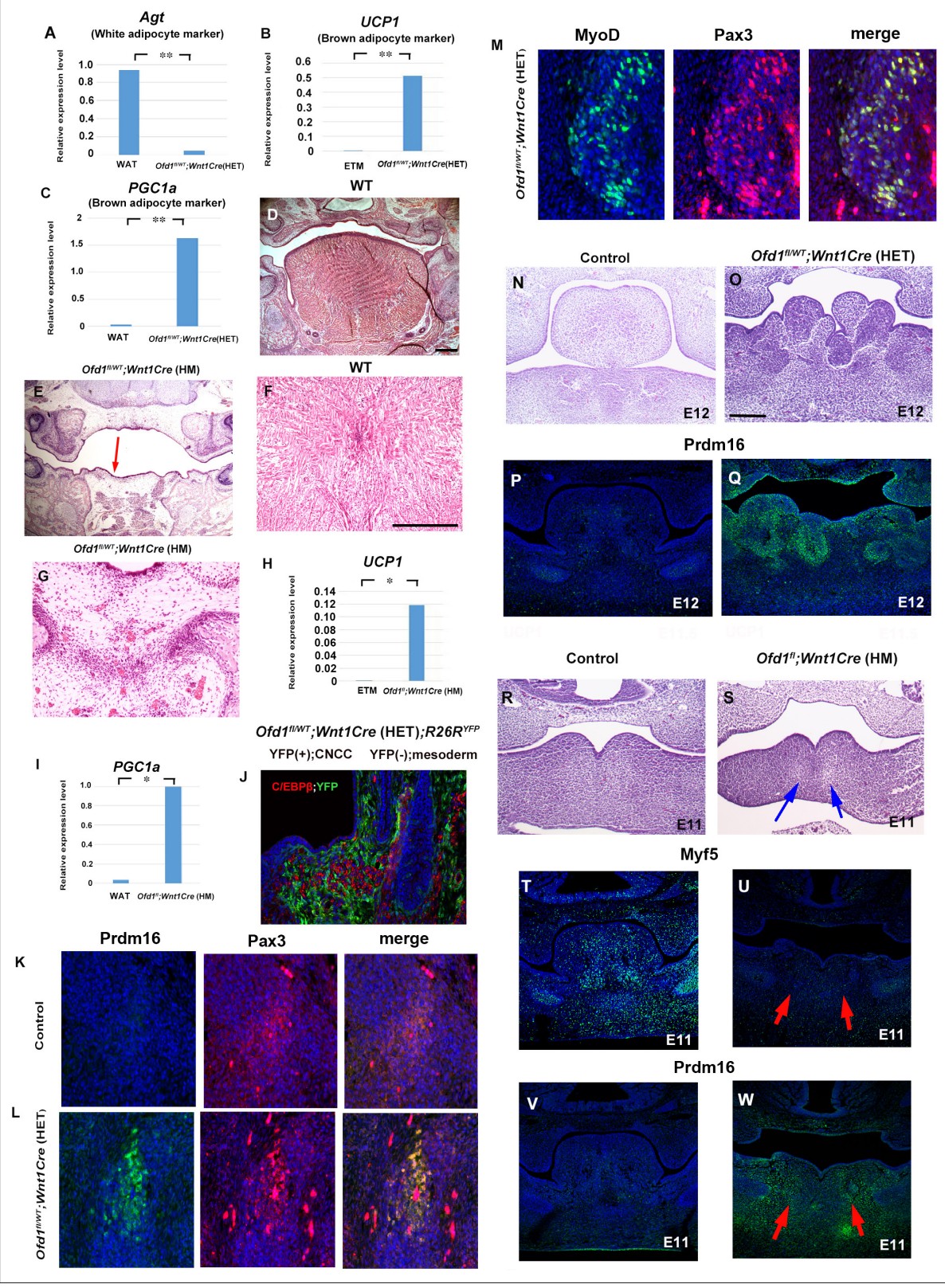

**Figure 2.** Sparse tissue in *Ofd1* mutant tongue. (**A–C**) Quantitative PCR (q-PCR) on mRNA isolated from ectopic sparse tissue, white adipose tissue (WAT), and embryonic tongue muscle (ETM). Expression of white adipose maker (*Agt*) was examined to understand whether the sparse tissue was white adipose maker. Only low-level expression of *Agt* in the sparse tissue compared with the WAT (**A**). Expression of brown adipose maker (*UCP1* and *PGC1a*) were examined to understand whether the sparse tissue was brown adipose maker. *UCP1* and *PGC1a* were expressed at high level in the

*Figure 2 continued on next page*

*Figure 2 continued*

sparse tissue, in comparison with those in the ETM (**B**) or adult WAT (**C**). **p<0.01. (**D–G**) Frontal sections showing histological images in wild-type (**D, F**) and *Ofd1^fl^;Wnt1Cre*(HM) (**E, G**). F and G are high magnification of D and E (indicated by blue arrow), respectively. (**H, I**) q-PCR on mRNA isolated from ectopic sparse tissue from *Ofd1^fl^;Wnt1Cre*(HM) mice, ETM and WAT. **p<0.01. (**J–M**) Frontal sections showing double immunohistochemistry of YFP and C/EBPβ in *Ofd1^fl/WT^;Wnt1Cre*(HET)*;R26R^YFP^* (**D**), Prdm16 and Pax3 in wild-type (**K**) and *Ofd1^fl/WT^;Wnt1Cre*(HET) (**L**), MyoD and Pax3 in *Ofd1^fl/WT^;Wnt1Cre*(HET) (**M**) at embryonic day (E) 13.5. (**N–Q**) Frontal sections showing histological images (**N, O**) and immunohistochemistry of Prdm16 (**P, Q**) in wild-type (**N, P**) and *Ofd1^fl/WT^;Wnt1Cre*(HET) mice at E12. (**R–W**) Frontal sections showing histological images (**R, S**) and immunohistochemistry of MyoD (**T, U**) and Prdm16 (**V, W**) in wild-type (**R, T, V**) and *Ofd1^fl^;Wnt1Cre*(HM) (**S, U, W**) mice at E11. Arrows indicating region showing sparse-like tissues. Scale bars: 200 μm (D, E, N, O, R, S), 100 μm (F, G).

The online version of this article includes the following source data and figure supplement(s) for figure 2:

**Source data 1.** qPCR; Agt, UCP1 and PGC1a.

**Figure supplement 1.** Quantitative PCR (q-PCR) and immunohistochemistry analysis (muscle and white adipose) on *Ofd1^fl/WT^;Wnt1Cre*(HET) mice.

**Figure supplement 1—source data 1.** qPCR; Myogenin and Trim13.

**Figure supplement 2.** Quantitative PCR (q-PCR) and immunohistochemistry analysis (brown adipose) on *Ofd1^fl/WT^;Wnt1Cre*(HET) mice.

**Figure supplement 2—source data 1.** qPCR; UCP1 and Adiponectin.

**Figure supplement 3.** Artifact and adipose tissue in wild-type tongue.

**Figure supplement 4.** Quantitative PCR (q-PCR) analysis (adipose) on *Ofd1^fl^;Wnt1Cre*(HM) mice.

**Figure supplement 4—source data 1.** qPCR; Agt, Trim14 and Prdm16.

---

Myogenic precursor cells confirmed by *Myf5* expression was observed within pharyngeal arch at the time (*Figure 3C*). However, *Myf5* expression could not be detected when *Ofd1* was deleted from all CNCC [*Ofd1^fl^;Wnt1Cre*(HM); *Figure 3D*]. It has been shown that *Osr2* is expressed in tongue connective tissue where CNCC form (*Figure 3—figure supplement 1*, *Lan et al., 2004*). In fact, at E13.5, we found that *Osr2Cre;R26R^LacZ^* mice showed LacZ expression in lamina propria (neural crest-derived region), but not in tongue muscle (*Figure 3—figure supplement 2A*). Furthermore, it has been shown that *Osr2-Cre* activation could not be detected in craniofacial region at E9.25 (*Lan et al., 2004*). At E10, *Osr2-Cre* activation was found in pharyngeal regions at E10 (*Figure 3E, E'*). At E11, both *Osr2*-expressing cells and tongue myoblast precursor cells were present in the mandibular process; however, *Osr2*-expressing cells did not interact with tongue myoblast precursor cells at the stage, due to the significant distance between *Osr2*-expressing cells and tongue myoblast precursor cells (confirmed by *MyoD* or *Myf5*) (*Figure 3F and G*, *Figure 3—figure supplement 2B–G*). Thus, *Osr2*-expressing cells likely interact with mesoderm-derived cells only within the pharyngeal regions at E10 (*Figure 3H*). Therefore, we generated *Ofd1* mutant mice using *Osr2-Cre* to understand whether interaction between CNCC and mesoderm-derived cells is occurred within pharyngeal arches at E10. Indeed, ectopic sparse tissue was observed in *Ofd1^fl^;Osr2Cre*(HM) mice (*Figure 3J*). To further confirm this, we generated *Ofd1* mutant mice using *Sox10-Cre*. *Sox10-Cre* has also been used for gene deletion from CNCC. However, *Sox-10Cre* activation starts in the middle of CNCC migration, while *Wnt1-Cre* is activated before their migration (*Hari et al., 2012*). Unlike *Wnt1-Cre*, *Sox10-Cre* is not activated immediately after start of migration. If interaction between CNCC and mesoderm-derived cells is crucial before CNCC migration for mesoderm-derived cell differentiation, tongue phenotype would not be present in *Ofd1* mutant generated by *Sox10-Cre*. If interaction between CNCC and mesoderm-derived cells is crucial within mandibular processes (after completion of their migration), tongue phenotypes would be same between *Ofd1* mutant generated by *Wnt1-Cre* and *Sox10-Cre*. If interaction between CNCC and mesoderm-derived cells is crucial during migration, tongue phenotypes would be partially rescued in *Ofd1* mutant generated by *Sox10-Cre*, since CNCC could interact with mesoderm-derived cells immediately after start of migration (while *Sox10-Cre* is inactivated). We compared *Ofd1^fl^;Sox10Cre*(HM) mice with *Ofd1^fl^;Wnt1Cre*(HM) mice, since tongue phenotypes were consistent in *Ofd1^fl^;Wnt1Cre*(HM) mice. We found the presence of slight muscle tissue in *Ofd1^fl^;Sox-10Cre*(HM) mice, which was never seen in tongue of *Ofd1^fl^;Wnt1Cre*(HM) mice (*Figure 3K and L*). Thus, interaction between CNCC and mesoderm-derived cells start immediately after migration. To further confirm the interaction between CNCC and mesoderm-derived cells for cell differentiation, in vitro assay was performed using mesenchymal cells obtained from *Wnt1Cre;R26R^YFP^* mice. These contained CNCC as YFP-posivive cells and mesoderm-derived cells as YFP-negative cells. Expression of muscle marker was observed in mesoderm-derived cells (YFP-negative cells) around CNCC

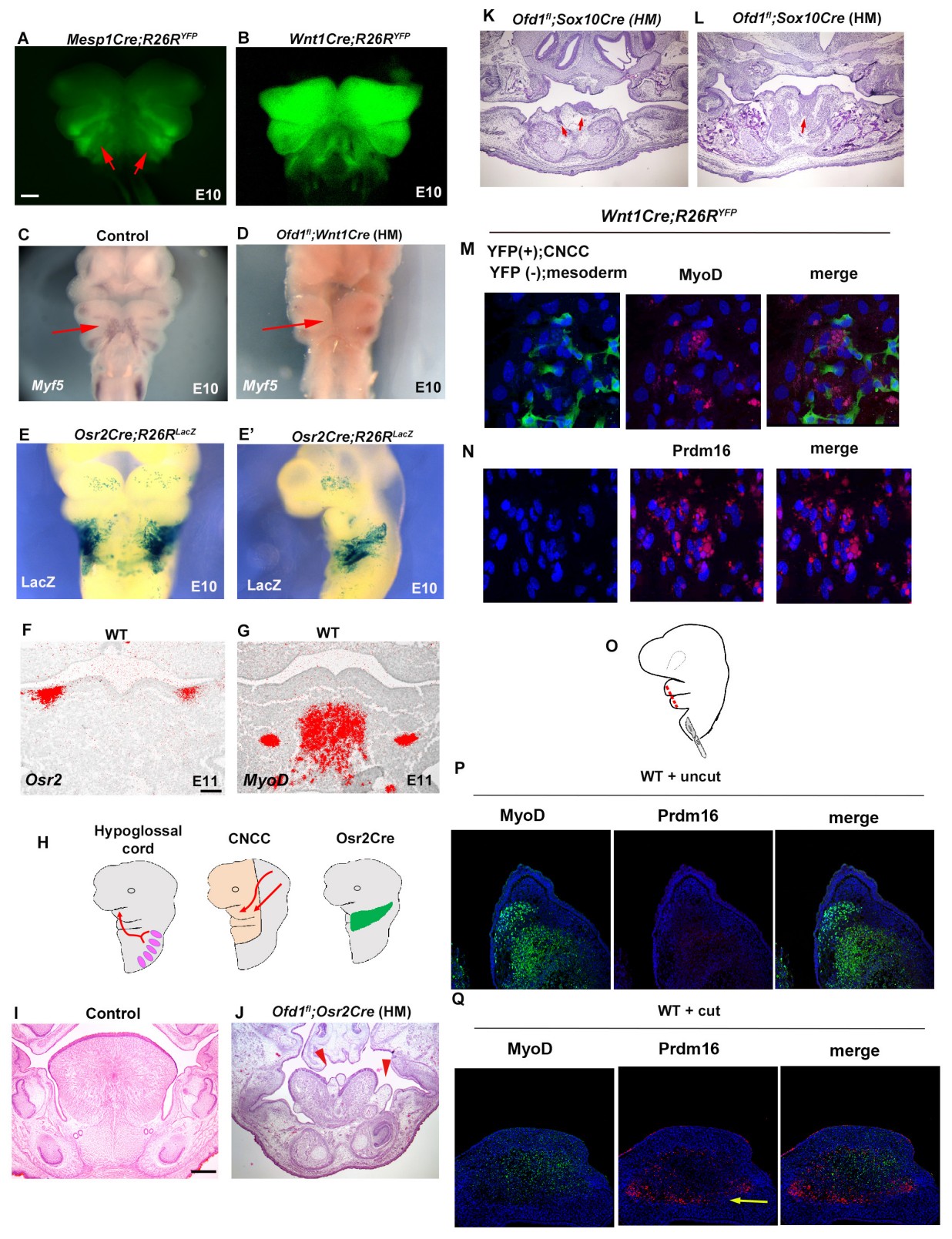

**Figure 3.** Interaction between cranial neural crest-derived cells (CNCC) and mesoderm-derived cells. (**A–D**) Frontal images of YFP expression (**A, B**) and *Myf5* expression of whole-mount in situ hybridization (**C, D**) in *Mesp1Cre;R26R^YFP* (**A**), *Wnt1Cre;R26R^YFP* (**B**), wild-type (**C**), and *Ofd1^fl;Wnt1Cre*(HM) (**D**) at embryonic day (E) 10. (**E, E'**) Frontal (**E**) and sagittal (**E'**) view of LacZ stained *Osr2Cre;R26R^LacZ* mice at E10. (**F, G**) Frontal sections showing in situ hybridization of *Osr2* (**F**) and *MyoD* (**G**) in wild-type at E11. (**H**) Schematic diagram showing hypoglossal cord (left), CNCC (middle), and *Osr2-Cre*

*Figure 3 continued on next page*

*Figure 3 continued*

expression domain (right). (**I–L**) Frontal sections showing histological images in wild-type (**I**), *Ofd1^fl^;Osr2Cre*(HM) (**L**), and *Ofd1^fl^;Sox10Cre*(HM) (K, anterior and L, middle) at E16.5 (**K, L**) and E18.5 (**I, J**). (**M**) Double immunohistochemistry of YFP and MyoD on cultured YFP-expressing CNCC accompanied by YFP-negative mesoderm-derived cells obtained from *Wnt1Cre;R26R^YFP^* mice. (**N**) Double immunohistochemistry of YFP and Prdm16 on cultured YFP-negative mesoderm-derived cells obtained from *Wnt1Cre;R26R^YFP^* mice. (**Q**) Schematic diagram showing lateral view of craniofacial region with incision (red line). (**P, Q**) Double immunohistochemistry of MyoD and Prdm16 in cultured wild-type tongue without incision (**P**) and with incision (**Q**). Arrowheads indicating Prdm16-positive cells (**Q**).The presence of adipose tissue in explants with incisions, n=4/8; and without incision, n=0/8. Scale bars: 500μm (A–E'), 200 μm (I, J, K, L), 100 μm (F, G).

The online version of this article includes the following figure supplement(s) for figure 3:

**Figure supplement 1.** Variation of *Osr2-Cre* activation.

**Figure supplement 2.** *Osr2* expression in developing tongue.

**Figure supplement 3.** Interaction between cranial neural crest-derived cells (CNCC) and mesoderm-derived cells.

**Figure supplement 4.** The cervical brown adipose tissue in *Ofd1^fl/WT^;Wnt1Cre*(HET) mice.

(YFP-posivive cells) (*Figure 3M*). Mesoderm-derived cells (YFP-negative cells) showed adipose marker expression, when they were isolated from CNCC (YFP-posivive cells) (*Figure 3N*). Furthermore, we used *Mesp1-Cre* to label mesoderm-derived cells with YFP through crossing *R26R^YFP^* mice for same in vitro assay (*Mesp1Cre;R26R^YFP^*). Expression of adipose marker was observed in only mesoderm-derived cells (GFP-positive cells) isolated from CNCC (GFP-negative cells), while these cells failed to show MyoD expression (*Figure 3—figure supplement 3*).

To further confirm whether differentiation of mesoderm-derived cells is relied on interaction between mesoderm-derived cells and CNCC within the pharyngeal region, we made several incisions into the pharyngeal region in wild-type mice to reduce the interaction, and then cultured the embryos. To minimize disturbance of their migration, punctiform incisions were made under the visualization of mesoderm-derived cells using *Mesp1Cre;R26R^YFP^* mice as wild-type mice (*Figure 3A and O*). Adipose tissue was observed in the tongue of wild-type with incisions (n=4/8, *Figure 3Q*), which was never seen in wild-type without incisions (n=0/8, *Figure 3P*). These results indicated that in normal tongue development, CNCC interact with mesoderm-derived cells within the pharyngeal region, which is essential for normal differentiation of mesoderm-derived cells into myoblasts. Aberrant interaction results in abnormal differentiation of mesoderm-derived cells into brown adipocytes. Thus, ectopic adipose formation due to aberrant differentiation of mesoderm-derived cells were caused by a lack of proper cell-cell interaction between CNCC and mesoderm-derived cells.

The supraclavicular and cervical brown tissues are present as the most anterior brown adipose tissues (*Figure 3—figure supplement 4A*, *Mo et al., 2017*). If ectopic adipose in *Ofd*1 mutant tongue was derived from the cervical brown adipose tissue due to mis-migration, the supraclavicular or cervical brown adipose tissue in *Ofd1* mutant mice would be shrinked or be connected to ectopic adipose in *Ofd1* mutant tongue. However, any shrinkage of these brown adipose tissue or connection between these brown adipose tissues and ectopic adipose could not be detected in *Ofd1* mutant mice (*Figure 3—figure supplement 4B–I*). Furthermore, ectopic brown adipose tissue in *Ofd1* mutant mice was found from E11.5 (*Figure 2W*), while the supraclavicular and cervical brown adipose tissue are known to be formed from E14.5 (*Sanchez-Gurmaches and Guertin, 2014*). Thus, ectopic brown adipose tissue found in *Ofd1* mutant mice is unlikely caused by mis-migration of the supraclavicular and cervical brown adipose tissue.

In addition to ectopic adipose tissues, normal muscle tissue was also present in *Ofd1^fl/WT^;Wnt-1Cre*(HET) tongue, indicating that some CNCC successfully interact and induced normal differentiation of mesoderm-derived cells into muscle progenitors, while other CNCC populations failed to elicit this. Thus, two types of CNCC (normal and abnormal CNCC) would appear to be present in *Ofd1^fl/WT^;Wnt1Cre*(HET) mice. This is most likely caused by X-inactivation, since *Ofd1* is located on the X-chromosome. To confirm this, we used *Hprt^GFP^* mice that have a GFP reporter expressed by Cre ribonuclease on the X-chromosome. We found that both GFP-positive cells [CNCC with paternal Ofd1 (+) activation] and GFP-negative cells [CNCC with maternal Ofd1 (+) activation] showed scattered pattern in *Wnt1Cre;Hprt^GFP^* mice made by crossing male *Hprt^GFP^* mice with female *Wnt1-Cre* mice (*Figure 4A*). In *Ofd1^fl^;Wnt1Cre*(HM);*Hprt^GFP^* mice generated through crossing female *Ofd1^fl/fl^* mice with male *Wnt1Cre;Hprt^YFP^* mice, CNCC with Ofd1 (-) chromosome activation [*Ofd1* mutant CNCC] could be

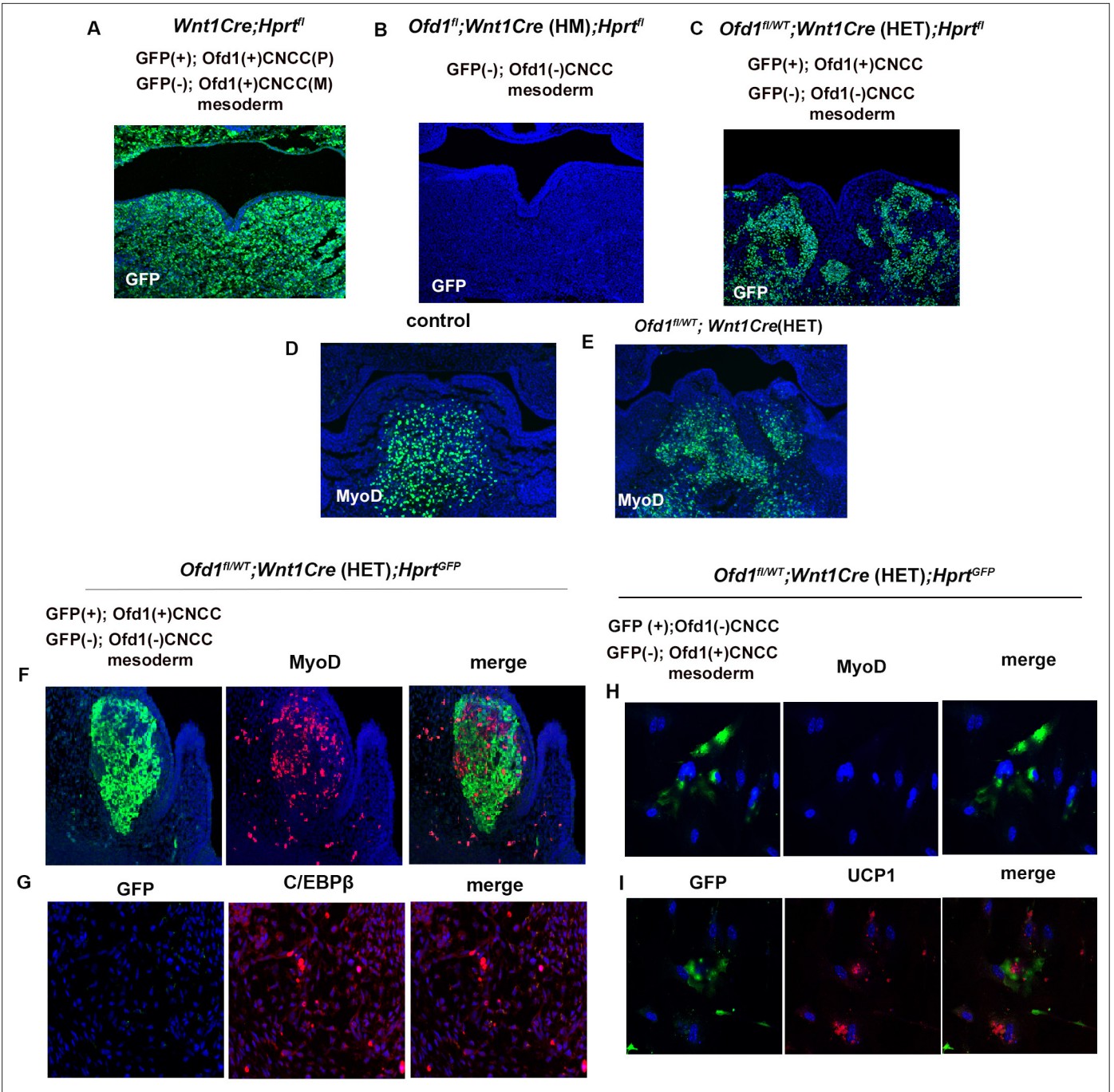

**Figure 4.** X-inactivation in *Ofd*1 mutant tongue. (**A–E**) Frontal sections showing immunohistochemistry of GFP (**A–C**) and MyoD (**D, E**) in *Wnt1Cre*(HET);*Hprt^fl^* (**A**), *Ofd1^fl^;Wnt1Cre*(HM);*Hprt^fl^* (**B**), *Ofd1^fl/WT^;Wnt1Cre(HET);Hprt^fl^* (**C**), wild-type (**D**), and *Ofd1^fl/WT^;Wnt1Cre*(HET) (**E**) mice at embryonic day (E) 11. P, paternal; M, maternal. (**F, G**) Double immunohistochemistry of GFP and MyoD (**F**), and GFP and C/EBPβ (**G**) in *Ofd1^fl/WT^;Wnt1Cre*(HET);*Hprt^fl^* mice. (**H, I**) Double immunohistochemistry of GFP and MyoD (**H**), and GFP and UCP1 (**I**) on cultured YFP-expressing Ofd1 (-) cranial neural crest-derived cells (CNCC), GFP-negative mesoderm-derived cells, and GFP-negative Ofd1 (+) CNCC obtained from *Ofd1^fl/WT^;Wnt1Cre*(HET);*Hprt^fl^* mice.

The online version of this article includes the following figure supplement(s) for figure 4:

**Figure supplement 1.** Cluster of *Ofd1* mutant cells in *Ofd1^fl/WT^;Wnt1Cre*(HET) mice.

detected as GFP-negative cells. All CNCC were found as GFP-negative cells in *Ofd1^fl^;Wnt1Cre*(HM);*Hprt^GFP^* mice (***Figure 4B***). Next, we generated *Ofd1^fl/WT^;Wnt1Cre*(HET);*Hprt^GFP^* mice which show cells with Ofd1 (+) chromosome activation [normal CNCC] and Ofd1 (-) chromosome activation [*Ofd1* mutant CNCC] as GFP-positive and -negative cells, respectively. In *Ofd1^fl/WT^;Wnt1Cre*(HET);*Hprt^GFP^*

mice, GFP-positive (normal CNCC) and -negative (*Ofd1* mutant CNCC) cells were observed as a mosaic pattern in the mandibular processes, indicating that both normal and *Ofd1* mutant CNCC were present in *Ofd1*$^{fl/WT}$;*Wnt1Cre*(HET) mice (**Figure 4C**, **Figure 4—figure supplement 1**). In wild-type mice, Myo-D was observed as a single domain (**Figure 4D**); however, *Ofd1*$^{fl/WT}$;*Wnt1Cre*(HET) mice showed a mosaic pattern of MyoD-positive domains (**Figure 4E**) as *Ofd1*$^{fl/WT}$;*Wnt1Cre*(HET);*Hprt*$^{GFP}$ mice showed (**Figure 4C**). MyoD-positive domains were found only around GFP-positive cells (normal CNCC) in *Ofd1*$^{fl/WT}$;*Wnt1Cre*(HET);*Hprt*$^{GFP}$ mice (**Figure 4F**). Adipose marker expression was found in the region where GFP-positive (normal CNCC) was absent (**Figure 4G**). To further confirm whether adipocyte formation occurs when mesoderm-derived cells contact with *Ofd1* mutant CNCC, in vitro assay was performed. For that experiment, we generated *Ofd1*$^{fl/WT}$;*Wnt1Cre*(HET);*Hprt*$^{GFP}$ mice to detect *Ofd1* mutant CNCC as GFP-positive cells. UCP1 expression was observed around GFP-positive cells (*Ofd1* mutant cells), while MyoD expression could not be observed in these cells (**Figure 4H and I**). Thus, *Ofd1* mutation in CNCC disrupts proper interactions between CNCC and mesoderm-derived cells, and subsequent differentiation of mesoderm-derived cells. These findings suggested that, in *Ofd1*$^{fl/WT}$;*Wnt1Cre*(HET) mice, CNCC displaying inactivation of the X-chromosome with *Ofd1* mutation induced normal differentiation of surrounding mesoderm-derived cells, while CNCC with inactivation of the normal X-chromosome failed to elicit this to surrounding mesoderm-derived cells. It is likely that CNCC in branchial arch region are randomly divided into two types of cell clusters (cells with and without *Ofd1*) due to the X-inactivation, since X-inactivation is based on random choice between the two X-chromosomes. In fact, mosaic pattern of GFP-positive (normal cells) and -negative (*Ofd1* mutant cell) cells varied widely in *Ofd1*$^{fl/WT}$;*Wnt1Cre*(HET);*Hprt*$^{GFP}$ mice (**Figure 4C**). This might lead to the phenotypic tongue variation seen in *Ofd1*$^{fl/WT}$;*Wnt1Cre*(HET) mice. Thus, the interaction between CNCC and mesoderm-derived cells is under control of X-inactivation.

*Ptch1* (major mediators of Hh signaling) was broadly expressed in the wild-type pharyngeal region, but not in the *Ofd1*$^{fl}$;*Wnt1Cre*(HM), suggesting that *Ofd1* mutation from all CNCC led to a complete absence of Hh signal activation in branchial arch region (**Figure 5A and B**). On the other hand, in *Ofd1*$^{fl/WT}$;*Wnt1Cre*(HET) mice, a mosaic pattern of *Ptch1*-negative and -positive domains were observed, which varied between *Ofd1*$^{fl/WT}$;*Wnt1Cre*(HET) mice (**Figure 5C**). These are also probably caused by X-inactivation. In fact, the pattern of *Ptch1*-negative and -positive domains were overlapped with GFP-negative and -positive domains in *Ofd1*$^{fl/WT}$;*Wnt1Cre*(HET);*Hprt*$^{GFP}$ mice, respectively (**Figure 5D and E**). To examine whether Hh signaling in CNCC controls differentiation of mesodermal cell, we inhibited Hh signaling in cultured mesenchymal cells obtained from *Wnt1Cre*;*R26R*$^{YFP}$ mice. MyoD was observed in YFP (-) cells contacted with CNCC [YFP (+) cells] in control group (**Figure 5F**). MyoD expression could not be seen in cyclopamine treated YFP (-) cells, although they contacted with CNCC (**Figure 5F**). Brown adipose marker expression could not be detected, when YFP (+) cells were present (**Figure 5G**). Brown adipose marker expression was observed in cyclopamine treated group, although CNCC [YFP (+) cells] were present (**Figure 5G**). To further understand whether ectopic brown adipose tissue formation was due to downregulation of the Hh signaling pathway in CNCC, we generated mice with conditional deletion of *Smo* (an essential mediator of Hh signaling activity) using *Osr2-Cre* (*Smo*$^{fl/fl}$;*Osr2Cre*) mice. Adipose-like tissue was observed in the tongue of *Smo*$^{fl/fl}$;*Osr2Cre* mice (**Figure 5I**). We already found ectopic brown adipose formation where tongue swelling is usually emerged, when *Ofd1* was deleted from all CNCC (**Figure 2D–I**, **Figure 2—figure supplement 4**, **Figure 2—figure supplement 4—source data 1**). The ectopic brown adipose was also observed in same region of mice with *Smo* deletion from all CNCC (*Smo*$^{fl/fl}$;*Wnt1Cre* mice; **Figure 5—figure supplement 1**, **Figure 5—figure supplement 1—source data 1**). Hh signaling is activated in primary cilia, and tongue anomalies are observed in ciliopathy patients. We also found alteration of primary cilia in CNCC of *Ofd1*$^{fl}$;*Wnt1Cre*(HM);*R26R*$^{YFP}$ mice (**Figure 5—figure supplement 2**). To confirm whether the disruption of primary cilia function induces tongue anomalies, we generated and examined mice with conditional deletion of *Ift88* (another primary cilia protein) using *Osr2-Cre* (*Ift88*$^{fl/fl}$;*Osr2Cre*). *Ift88*$^{fl/fl}$;*Osr2Cre* mice also exhibited same abnormal tongue as those in *Ofd1*$^{fl/WT}$;*Wnt1Cre*(HET) mice (**Figure 5J**). Thus, the interaction between CNCC and mesoderm-derived cells are reliant on Hh signaling in CNCC, which is essential for proper differentiation of mesoderm-derived cells.

Downregulation of Hh signaling was confirmed in cyclopamine treated cultured cells obtained from *Wnt1Cre*;*R26R*$^{YFP}$ mice (**Figure 6—figure supplement 1**). When we confirmed it, we noticed that in control sample, mesoderm-derived cells [YFP (-) cells] around CNCC showed weak Hh signaling

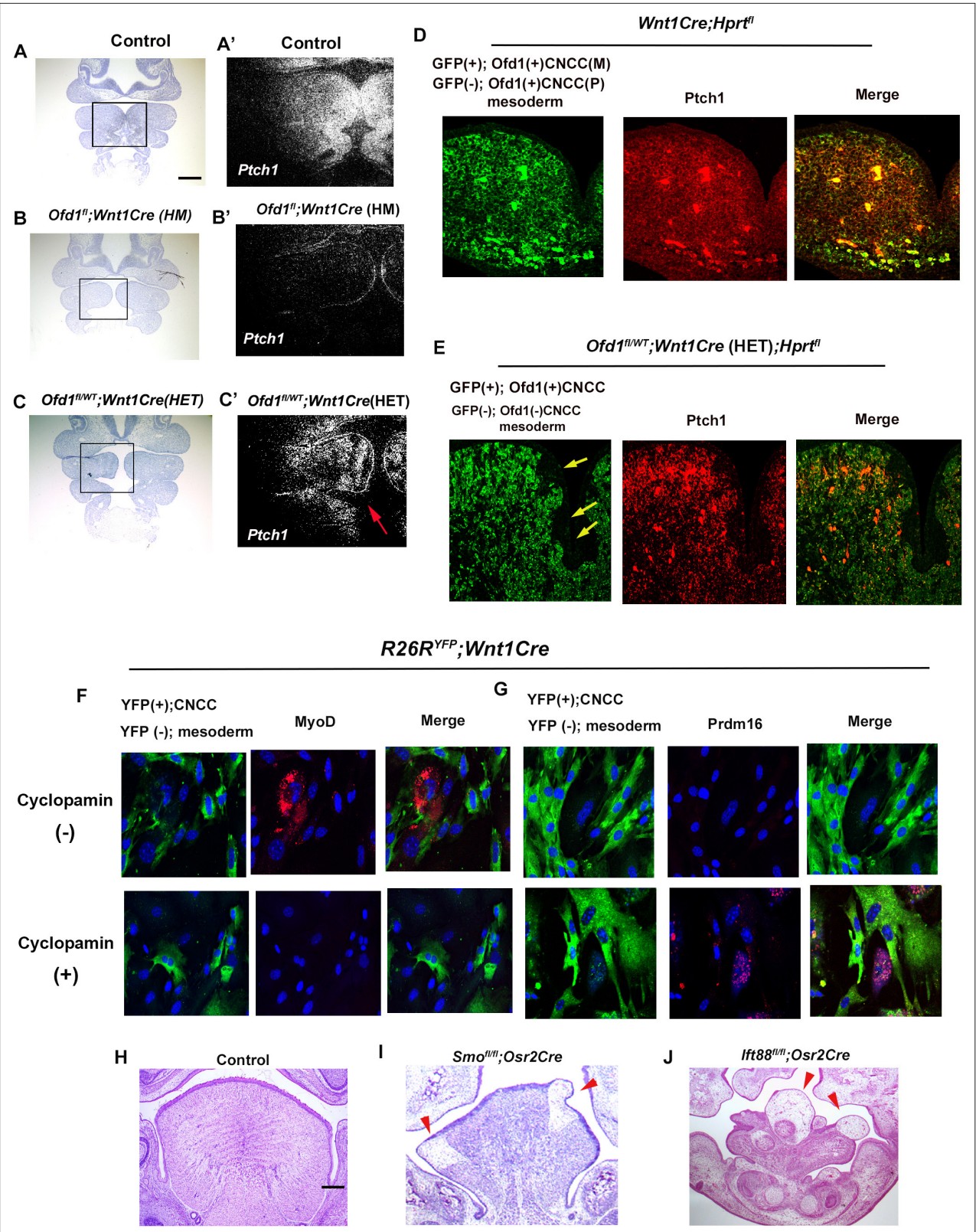

**Figure 5.** Hh signal in *Ofd*1 mutant tongue. (**A–C'**) Frontal sections showing in situ hybridization of *Ptch1* in wild-type (**A'**), *Ofd1^fl^;Wnt1Cre*(HM) (**B'**), and *Ofd1^fl/WT^;Wnt1Cre*(HET) (**C'**) at embryonic day (E) 10. A, B, C, bright field (counterstained section by hematoxylin) of A', B', C', respectively. A', B', C', high magnification of regions outlined by boxes in A, B, C, respectively. Arrow indicating mosaic *Ptch1* expression (**C'**). (**D, E**) Frontal sections showing double immunohistochemistry of GFP and Ptch1 in *Wnt1Cre;Hprt^fl^* (**D**) and *Ofd1^fl/WT^;Wnt1Cre*(HET)*;Hprt^fl^* (**E**) mice. Arrow indicating GFP-negative region (**E**). (**F,**

*Figure 5 continued on next page*

*Figure 5 continued*

**G**) Double immunohistochemistry of YFP and MyoD (**F**), and YFP and Prdm16 (**G**) on cultured YFP-expressing cranial neural crest-derived cells (CNCC) accompanied by YFP-negative mesoderm-derived cells obtained from *Wnt1Cre;R26R^YFP* mice with (lower panels) or without (upper panels) cyclopamine. (**H–J**) Frontal sections showing histology of wild-type (**F**), *Smo^fl/fl^;Osr2Cre* (**G**), and *Ift88^fl/fl^;Osr2Cre* (**H**) at E18.5. Arrowheads indicating ectopic sparse tissue. Scale bars: 100 µm (A-C'), 200 µm (H-J).

The online version of this article includes the following source data and figure supplement(s) for figure 5:

**Figure supplement 1.** Adipose in *Smo* mutant mice.

**Figure supplement 1—source data 1.** qPCR; Myogenin, Cidea, C/EBPb, Prdm16, UCP1, PGC1a, Agt, Trim14 and Cidea.

**Figure supplement 2.** Acetylated α-tubulin in cranial neural crest-derived cells (CNCC) during tongue development.

(*Figure 6A*). Ptch1 expression could not be detected in mesoderm-derived cells [YFP (-) cells] isolated from CNCC (*Figure 6B*). It is possible that CNCC with Hh signaling induce Hh signaling in mesoderm-derived cells. In fact, in common with these in vitro assay, Ptch1 expression was observed in YFP(-) cells of *Wnt1Cre;R26R^YFP* mice (*Figure 6C*). Furthermore, whole-mount in situ hybridization exhibited that Hh signaling was activated in the region where Myf5 was expressed (*Figure 6D and E*). We already confirmed that Hh signaling in CNCC was downregulated due to *Ofd1* mutation in CNCC. In addition to CNCC, Hh signaling activity was diminished where Mfy5 was usually expressed in *Ofd1^fl^;Wnt-1Cre*(HM) mice (*Figure 6F and G*). It has been shown that Hh signaling in mesodermal cells determine their cell fate for adipocyte or myoblasts (*Nosavanh et al., 2015*; *Norris et al., 2023*). To confirm the role of Hh signaling in mesoderm-derived cells in tongue development, Hh signaling activity was elicited by SAG in mesoderm-derived cells without contacting with CNCC. MyoD expression was observed in mesoderm-derived cells, even though CNCC was absent (*Figure 6H*).

Hh signaling has been shown to be involved in regulation of apoptosis and cell proliferation in mandibular processes (*Jeong et al., 2004*; *Millington et al., 2017*). However, significant changes of apoptotic cell number or cell proliferation could not be detected in *Ofd1^fl^;Wnt1Cre*(HM) mice (*Figure 6—figure supplement 2*).

## Tongue swelling formation

*Ofd1^fl^;Wnt1Cre*(HM) mice exhibit aglossia, indicating that an early tongue swelling does not occur when *Ofd1* is deleted from all CNCC (*Figure 1B*, *Okuhara et al., 2019*). *Ofd1^fl^;Wnt1Cre*(HM) mice also showed adipose tissue in the region where the tongue usually emerged (*Figure 2E and G*, *Figure 2—figure supplement 4*, *Figure 2—figure supplement 4—source data 1*). The absence of a tongue swelling in *Ofd1^fl^;Wnt1Cre*(HM) mice suggested that ectopic brown adipocytes could not develop a tongue swelling. Unlike *Ofd1^fl^;Wnt1Cre*(HM) mice, tongue was present in *Ofd1^fl/WT^;Wnt1Cre*(HET) mice, and abnormal deep clefts originating from the tongue root were often observed, while adipose tissue was seen at the bottom of these clefts (*Figure 7A–C*, *Figure 1—figure supplement 1*). Sagittal sections of the cleft part of the tongue also showed that the adipose tissue was present at the most posterior part of the cleft (*Figure 7D*). Fate mapping analysis using DiD labeling (injection where the early tongue emerges = TE region; *Figure 7E*) confirmed that the tongue develops along the antero-posterior axis (*Figure 7E–E''*). This suggested the possibility that the presence of adipose disturbs the formation of the early tongue swelling, which lead to the cleft formation in *Ofd1^fl/WT^;Wnt1Cre*(HET) mice, since ectopic adipose could not form tongue swelling in *Ofd1^fl^;Wnt1Cre*(HM) mice. To confirm this hypothesis, myoblasts and CNCC were replaced with adipose tissue in wild-type mandible at E11.0 where the tongue emerges, and cultured them (*Figure 7F*). The replacement was performed only restricted area within the region tongue swelling usually emerges. The cleft-like structures were observed in the cultured tongue, which was never seen in cultured tongue without replacement (*Figure 7G–H'*). Thus, cleft is occurred when adipose tissue was present where CNCC/myoblasts are usually located before emergence of tongue swelling. These also suggested that tongue morphology was determined by which tissue first reaches the region where the tongue swelling emerges. Tongue swelling occur when CNCC/myoblasts reach first, while it is inhibited when adipocytes reach first.

We already concluded that ectopic adipose tissue could not form a tongue swelling; however, adipose tissue was often seen within the tongue swelling of *Ofd1^fl/WT^;Wnt1Cre*(HET) mice (*Figure 8A and B*). Our findings indicated that early formation of the tongue swelling requires the presence of CNCC and mesoderm-derived cells. We often found that adipose tissue was present in the dorsum

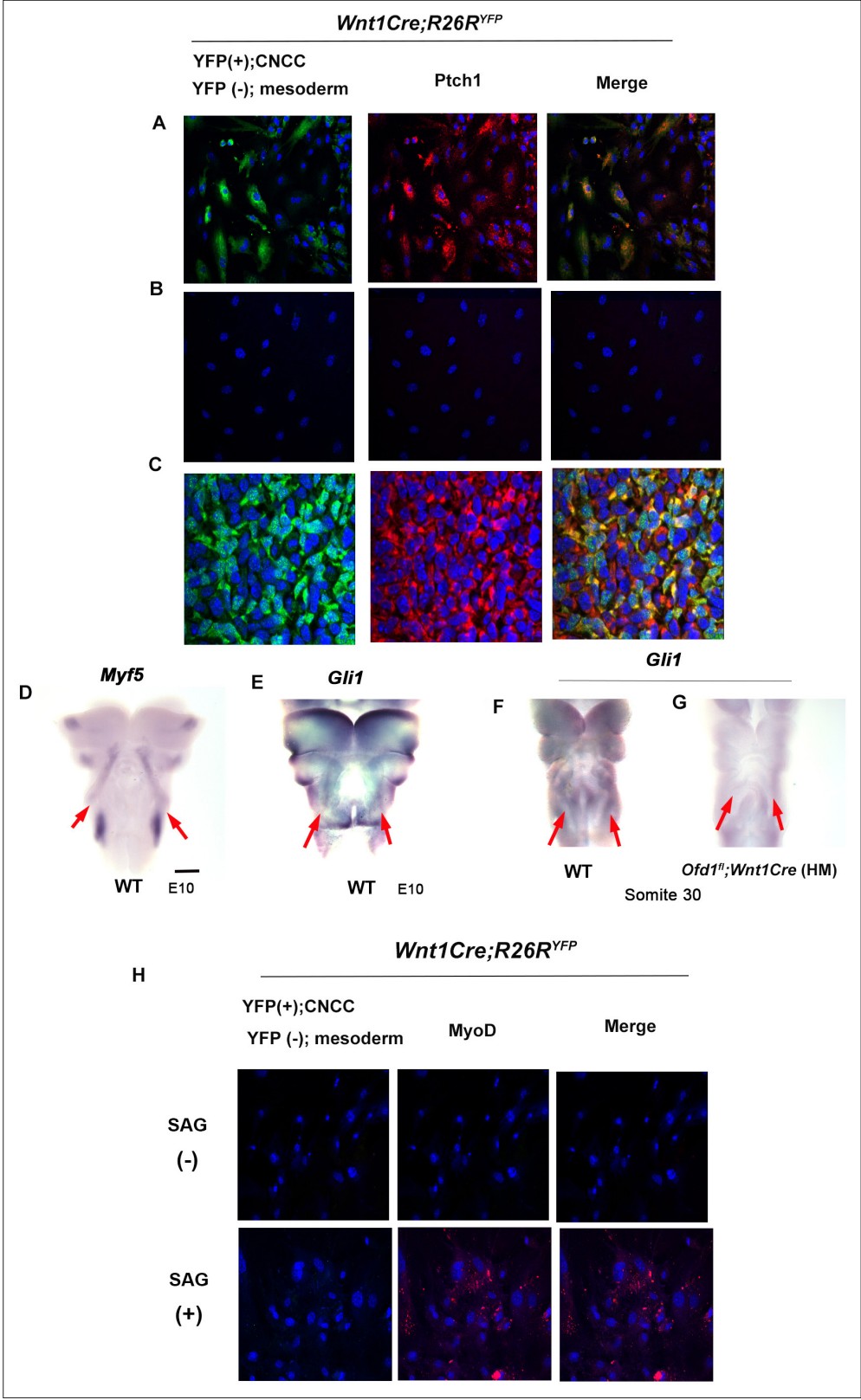

**Figure 6.** Hh signal in mesoderm-derived cells. (**A, B**) Double immunohistochemistry of YFP and Ptch1 on cultured YFP-expressing cranial neural crest-derived cells (CNCC) accompanied by YFP-negative mesoderm-derived cells (**A**) and only YFP-negative mesoderm-derived cells (**B**) obtained from *Wnt1Cre;R26R*^YFP mice. (**C**) Double immunohistochemistry of YFP and Ptch1 in *Wnt1Cre;R26R*^YFP mice. (**D–G**) Frontal images of whole-mount in situ

*Figure 6 continued on next page*

*Figure 6 continued*

hybridization of *Myf5* (**D**) and *Gli1* (**E–G**) in wild-type (**D–F**) and *Ofd1^fl^;Wnt1Cre*(HM) (**G**) at embryonic day (E) 10. (**H**) Double immunohistochemistry of YFP and MyoD on cultured YFP-negative mesoderm-derived cells obtained from *Wnt1Cre;R26R^YFP^* mice with (lower panels) or without (upper panels) SAG. Scale bars: 500 µm (D-G).

The online version of this article includes the following source data and figure supplement(s) for figure 6:

**Figure supplement 1.** Cyclopamine treated cranial neural crest-derived cells (CNCC) and mesoderm-derived cells.

**Figure supplement 2.** Apoptosis and cell proliferation in *Ofd1* mutant tongue.

**Figure supplement 2—source data 1.** Percentage of Positive cells: Caspase-3 and Ki67.

region of the emerging tongue swelling at E12.5, when the tongue swelling emerged in *Ofd1^fl/WT^;Wnt-1Cre*(HET) mice (*Figure 8C and C'*). It is possible that myogenic progenitors and CNCC push adipocytes up during early formation of the tongue swelling. To confirm this, we examined the formation of tongue papillae. In wild-type mice, the tongue papillae initiate through epithelial-mesenchymal interactions during emergence of the early tongue swelling. It is unlikely that adipocytes induce tongue papillae epithelium through epithelial-mesenchymal interaction. Therefore, the presence or absence of tongue papillae would be an indicator of whether adipocytes are located underneath the tongue epithelium during emergence of the tongue swelling. The epithelium and lamina propria overlying the tongue muscle exhibited tongue papillae-like structures in *Ofd1^fl/WT^;Wnt1Cre*(HET) mice (*Figure 8E and E'*). However, only a thin epithelium was observed in the region overlying the ectopic adipose tissue (*Figure 8E and E''*). The tongue papillae marker *Krt1-5* (*Jonker et al., 2004*) was expressed in the epithelium overlying muscle tissue with lamina propria only, and not in the overlying ectopic adipose tissue (*Figure 8F and G*). The tongue papillae was also observed in *Ofd1* mutant using *Sox10-Cre* (*Figure 3K and L*), which further confirm the presence of muscle in *Ofd1^fl^;Sox10Cre*(HM) mice. Thus, the presence of ectopic adipose tissue underneath the tongue epithelium led to a lack of tongue papillae. To further confirm this finding, we implanted adipose tissue on CNCC and myoblasts in wild-type mandibular explants at E11.0, and cultured them (*Figure 8H*). The adipose tissue was found on the dorsum region after tongue swelling (*Figure 8I and J*). Thus, adipocytes could enter the tongue swelling when adipocytes are located above CNCC and/or myoblasts. These also confirmed that tongue morphology was determined by which tissue first reaches the region where the tongue swelling emerges. Adipose could enter to tongue primordia when adipocytes reach after CNCC/myoblasts. Adipose tissue often goes down to the floor of the mandible from the dorsum of the tongue during emergence of tongue swelling (*Figure 8L*; see *Figure 8A and B*).

## Ectopic adipose in OFD1 patient

We next examined the tongue of an individual with *OFD1* G138S missense mutation. Several protrusions were observed in the tongue, which displayed an abnormally smooth surface in some regions, while others exhibited a normal dorsum surface (*Figure 9A and B*). Similar to *Ofd1^fl/WT^;Wnt1Cre*(HET) mice, sparse tissue was observed in the tongue (*Figure 9E and F*). PCR analysis (comparing cultured human skeletal muscle myoblasts or white adipocytes) and immunohistochemistry indicated that the sparse tissue observed in the tongue of the OFD1 subject was brown adipose (*Figure 9I–N*, *Figure 9—source data 1*, *Figure 9—figure supplement 1*, *Figure 9—figure supplement 1—source data 1*). Furthermore, no obvious tongue papillae were detected in the protruded tissue (*Figure 9P*), and expression of KRT1-5 could not be detected in epithelium of the protruded tissue (*Figure 9R*). Thus, our findings in mice likely occur in patient tongue formation.

## Tongue frenum formation

The tongue consists of a main body and tongue frenum. The tongue frenum is often absent in *Ofd1^fl/WT^;Wnt1Cre*(HET) mice (see *Figure 1H*). The tongue frenum is a thin flap of mucous membrane, and the tissue underneath the epithelium consists of CNCC, but not myogenic cells (*Figure 10A and B*). The tongue frenum is already formed in wild-type mice at E12 when the tongue swelling becomes obvious (*Figure 10—figure supplement 1A*), suggesting that the tongue frenum likely forms during emergence of the tongue swelling. It has been shown that tongue frenum is absent in *Lgr5* and *Tbx22* mutant mice (*Morita et al., 2004*; *Pauws et al., 2009*). We found restricted expression pattern of *Lgr5*

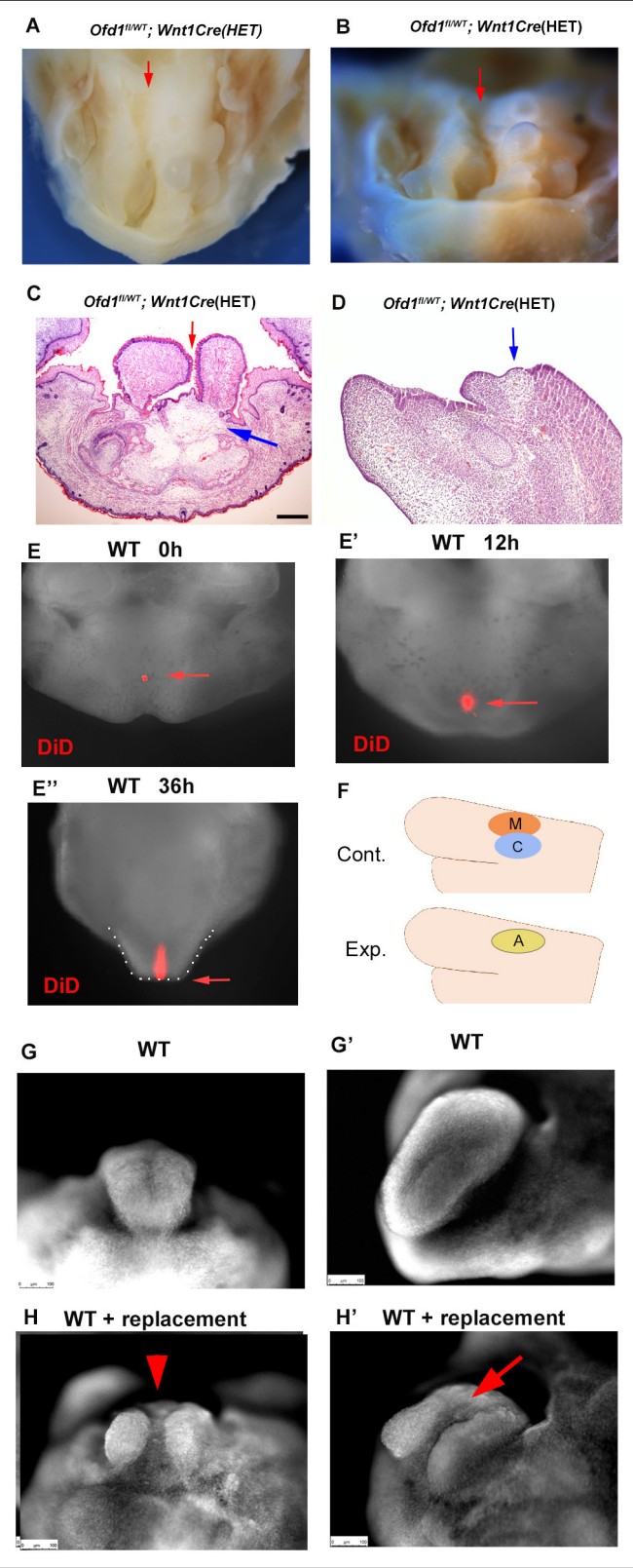

**Figure 7.** Clefts in *Ofd1* mutants. (**A, B**) Oral (**M**) and frontal (**N**) view of *Ofd1^fl/WT^;Wnt1Cre*(HET) mice. Arrows indicating cleft. (**C, D**) Frontal (**C**) and sagittal (**D**) sections showing histological images. Blue and red arrows indicating sparse tissue and cleft, respectively. (**E–E''**) Nuclear fluorescent image showing oral view of wild-type mandible with DiD at embryonic day (E) 11.5 before culture (**E**), 12 hr (**E'**), and 36 hr (**E''**) after culture. Arrow

*Figure 7 continued on next page*

*Figure 7 continued*

indicating DiD labeled cells. Tongue-like structure is outlined by white dots (**E″**). (**F**) Schematic diagram showing sagittal view of mandible exhibiting replacement of cranial neural crest-derived cells (CNCC) (C, blue circle) and myoblasts (M, red circle) into adipose tissue (A, yellow circle). (**G–H′**) Nuclear fluorescent image showing frontal (**G, H**) and oral (**G′, H′**) view of wild-type mandible without (**G, G′**) and with (**H, H′**) replacement. Scale bars: 200 µm (C, D).

and *Tbx22* in wild-type mandibular processes at E11.5 (*Figure 10C and D*). *Lgr5* expression could not be detected in the mandibular processes at E10.5 (*Figure 10E*, *Figure 10—figure supplement 1B*). This indicated that the region showing *Lgr5* and *Tbx22* at E11.5 are probably involved in tongue frenum formation. Although the tongue frenum is located at the midline, *Tbx22* showed no expression in the midline of the mandibular processes at E11.5 (*Figure 10D*). In order to determine whether

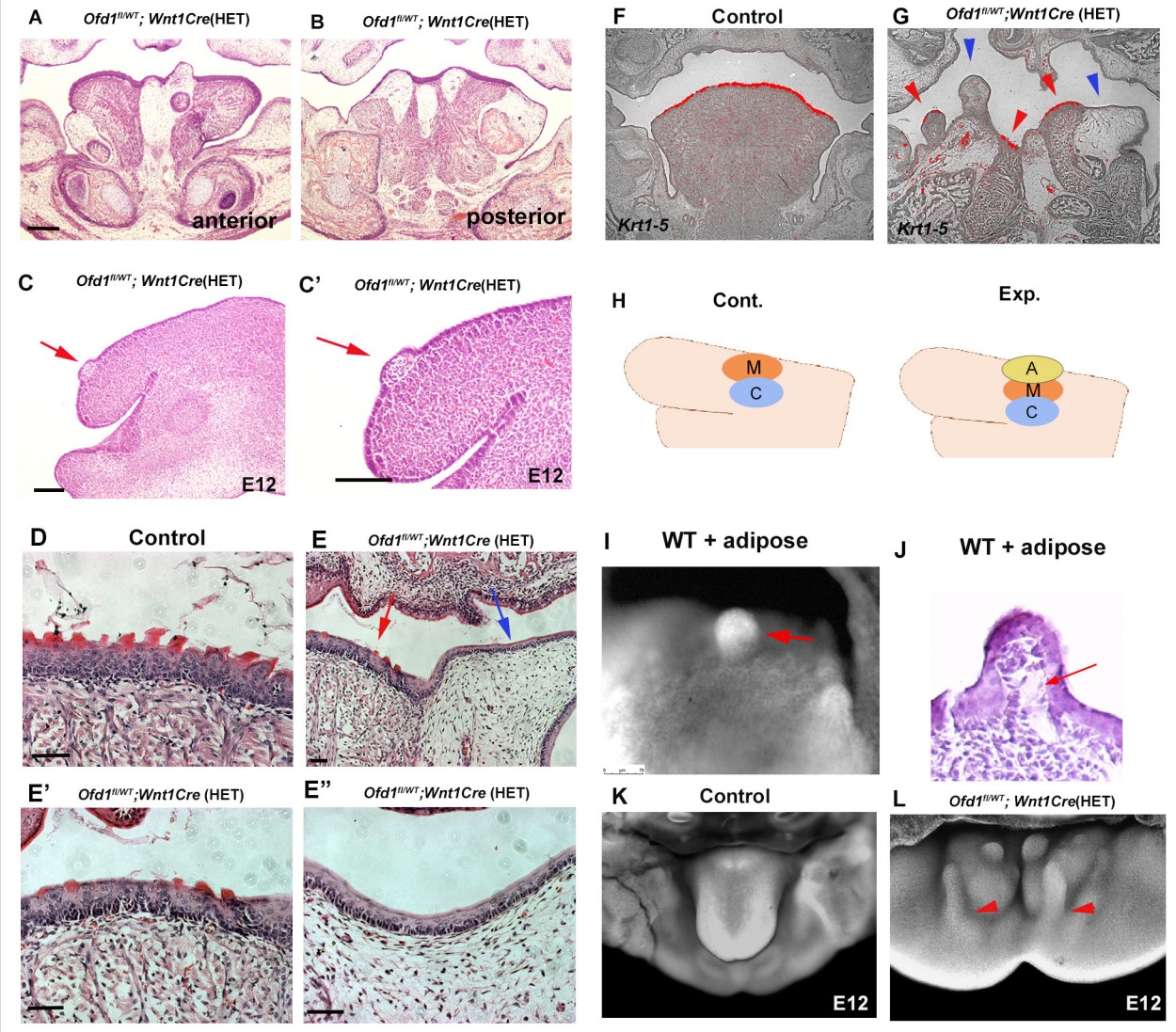

**Figure 8.** Adipose tissue in *Ofd1* mutant tongue. (**A, B, C–E″**) Frontal (**A, B, D–E″**) and sagittal (**C, C′**) sections showing histological images in wild-type (**D**), and *Ofd1^{fl/WT}*;*Wnt1Cre*(HET) (**A–C′, E–E″**). E′ and E″ are high magnification of E. (**F, G**) Frontal sections showing in situ hybridization of *Krt1-5* in wild-type (**F**) and *Ofd1^{fl/WT}*;*Wnt1Cre*(HET) (**G**). (**H**) Schematic diagram showing sagittal view of mandible exhibiting graft of adipose tissue (A, yellow circle), cranial neural crest-derived cells (CNCC) (C, blue circle) and myoblasts (M, red circle). (**I**) Nuclear fluorescent image showing dorsum of tongue in cultured mandible with grafting. (**J**) Histological images of dorsum of tongue in cultured mandible with grafting. (**K, L**) Nuclear fluorescent image showing oral view of developing mandibles in wild-type (**K**) and *Ofd1^{fl/WT}*;*Wnt1Cre*(HET) (**L**). Arrowheads indicating protrusion into the floor of mandible from the dorsum surface of tongue swelling (**L**). Scale bars: 200 µm (A, B, F, G), 100 µm (C-E″).

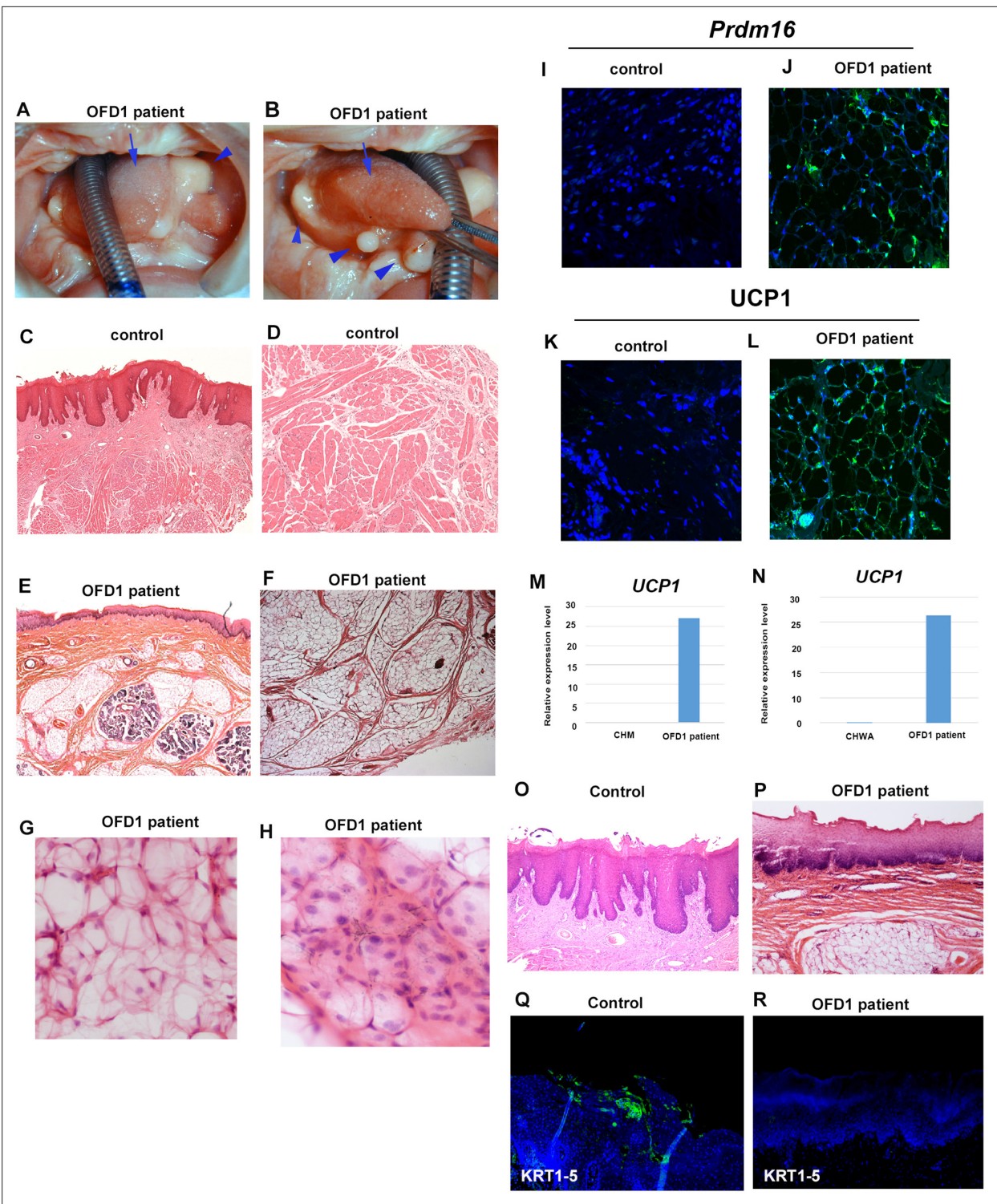

**Figure 9.** Tongue in OFD1 patient. (**A, B**) Tongue in OFD1 patient (G138S missense mutation). Arrowheads and arrows indicating protruded tissue and normal tongue, respectively. (**C–L, O–R**) Sections showing histological images (**C–H, O, P**) and immunohistochemistry of Prdm16 (**I, J**), UCP1 (**K, L**), and KRT1-5 (**Q, R**) on control human tongue (**C, D, O, Q**) and OFD1 patient tongue (**E–H, J, L, P, R**). (**M, N**) Quantitative PCR (q-PCR) on mRNA isolated from ectopic sparse tissue from OFD1 patient tongue, cultured human skeletal muscle myoblasts (CHM), and cultured human white adipocyte (CHWA).

The online version of this article includes the following source data and figure supplement(s) for figure 9:

**Source data 1.** qPCR; UCP1.

**Figure supplement 1.** Quantitative PCR (q-PCR) analysis on OFD1 patient tongue.

**Figure supplement 1—source data 1.** qPCR; CIDEA, PGC1a, PRDM16 and CIDEA.

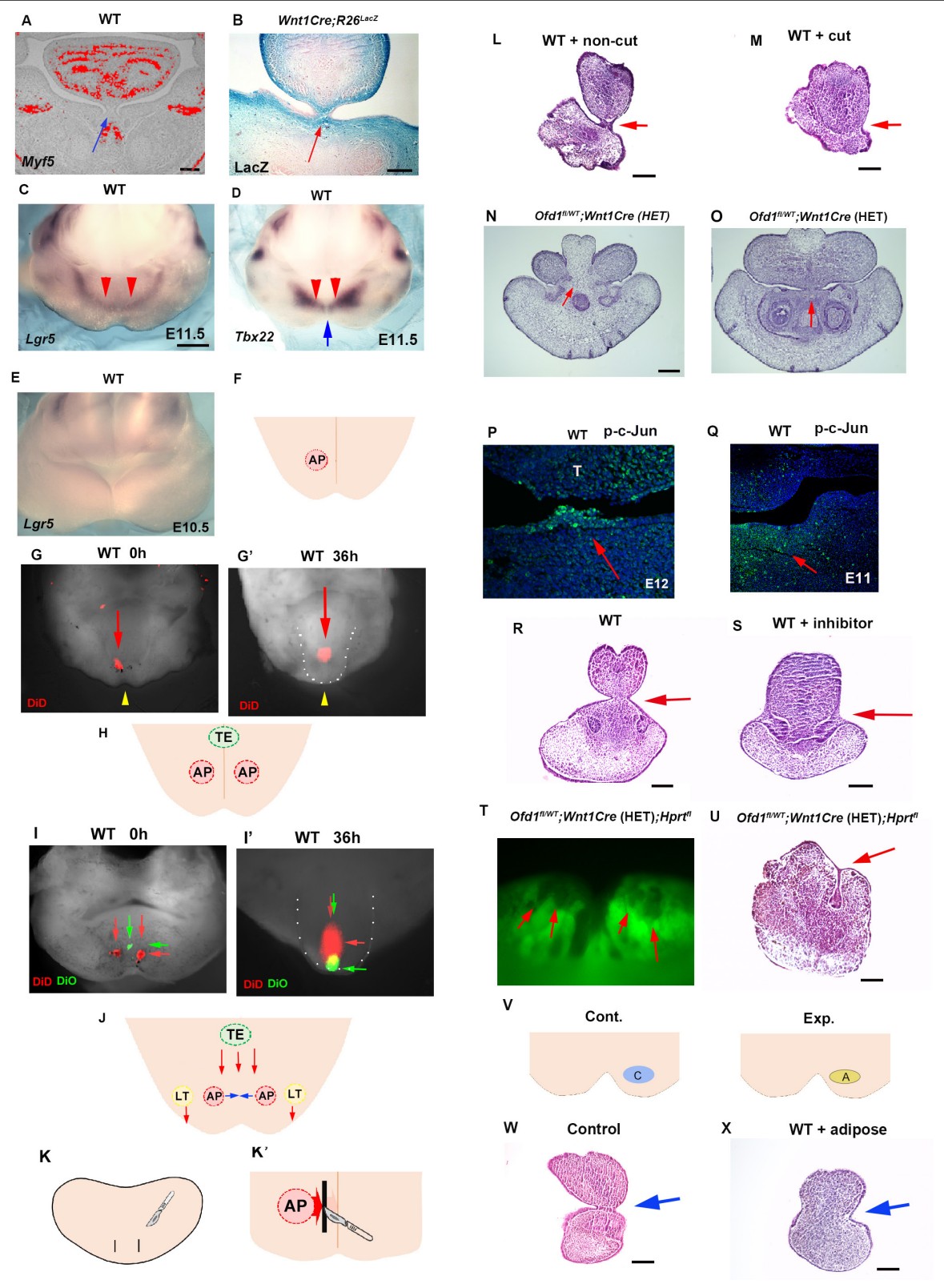

**Figure 10.** Tongue frenum in *Ofd1* mutant mice. (**A, B**) Frontal sections showing in situ hybridization of *Myf5* (**A**) and LacZ staining (**B**) in wild-type (**A**) and *WntCre;R26R^LacZ^* (**B**). Arrows indicating tongue frenum region. (**C–E**) Oral view of whole-mount in situ hybridization of *Lgr5* (**C, E**) and *Tbx22* (**D**) in wild-type at embryonic day (E) 11.5 (**C, D**) and E10.5 (**E**). Red and blue arrows indicating anterior expression domain and midline region, respectively. (**F, H**) Schematic diagram showing oral view of mandible with AP (light red) and TE region (light green), and midline (orange line). (**G, G', I, I'**) Nuclear

*Figure 10 continued on next page*

*Figure 10 continued*

fluorescent image showing oral view of wild-type mandible before (**G, I**) and after (**G', I'**) culture. Red and green arrows indicating DiD and DiO, respectively. Yellow arrowheads indicating midline. (**G', I'**) Tongue-like structure was partially removed and presumptive tongue-like structure are outlined by white dots. (**J**) Schematic diagram showing oral view of mandible with movement of cells in AP (light red), LT (light yellow), and TE region (light green line). (**K, K'**) Schematic diagram showing oral view of mandible with incision (black line) and midline (orange line). (**L, M**) Frontal sections showing histological images of cultured jaw without incision (**L**) and with incision (**M**). Arrows indicating the tongue frenum region. Tongue frenum formation with, n=1/6 and without incision, n=8/10. (**N, O**) Frontal sections showing histological images in *Ofd1*$^{fl/WT}$*;Wnt1Cre*(HET) mice obtained from same mouse (**O**; more posterior site than N). (**P, Q**) Frontal sections showing immunohistochemistry of p-c-Jun in wild-type at E12 (**P**) and E11 (**Q**). Arrows indicating positive cells in the tongue frenum (**P**) and AP region (**Q**). (**R, S**) Frontal sections showing histological images of cultured jaw with (**S**) and without U73122 (**R**). Arrows indicating the tongue frenum region. Tongue frenum formation with U73122, n=0/4. (**T**) Oral view of mandible in *Ofd1*$^{fl/WT}$*;Wnt1Cre*(HET)*;Hprt*$^{fl}$ mice. Arrows indicating Gfp-negative domains around AP region before culture. (**U**) Frontal sections showing histological images of T after culture. Arrow indicating tongue frenum region. (**V**) Schematic diagram showing oral view of mandible with replacement (A, yellow circle=adipose, blue circle=cranial neural crest-derived cell [CNCC]). (**W, X**) Frontal sections showing histological images of cultured jaw without replacement (**W**) and with replacement (**X**). Arrows indicating tongue frenum region. Tongue frenum formation with replacement, n=1/5. Scale bars: 200 μm (A, B, L, M, N, O, R, S, U, W, X), 500 μm (C-E).

The online version of this article includes the following figure supplement(s) for figure 10:

**Figure supplement 1.** Tongue frenum formation in wild-type mice.

the anterior region of the *Lgr5/Tbx22* expression domain is linked to tongue frenum formation, fate mapping analysis using DiD was performed in the wild-type mandible. DiD was injected within the anterior part of *Lgr5/Tbx22*-expressing domain (AP region, ***Figure 10F and G***, ***Figure 10—figure supplement 1C–F***). DiD in the AP region moved to the midline region (***Figure 10G'***). Thus, cells at the AP region moved along the lingual-buccal axis and were then retained at the midline. Moreover, DiD labeled cells were located at tongue frenum region after culture (***Figure 10—figure supplement 1G***). When we injected DiO to more lateral regions of the *Lgr5/Tbx22*-expressing domain (LT region, ***Figure 10—figure supplement 1C***, ***Figure 10—figure supplement 1H***), cells labeled with DiO in the LT region moved along the anterior-posterior axis (***Figure 10—figure supplement 1H' and H''***). We already concluded that the TE region (the more posterior part of the mandibular processes from the AP regions at the midline) moved along the anterior-posterior axis (see ***Figure 7E–E''***). We then injected DiO at the TE region, and DiD at both left and right AP regions (***Figure 10H and I***). Cells labeled with DiO (TE region) moved along the anterior-posterior axis and overtook cells with DiD (AP region) which moved across the lingual-buccal axis and merged at the midline (***Figure 10I'***). Thus, distinct axis of tissue formation is present to form tongue swelling and frenum (***Figure 10J***). To confirm whether cells in AP region form tongue frenum, we made incisions a lingual-buccal axis between the midline and AP region in wild-type mandibular processes to disturb tissue formation from the AP region to the midline, and cultured embryos (***Figure 10K and K'***). No tongue frenum was observed in cultured mandibles with incision (n=1/6, ***Figure 10M***), while the tongue frenum was present in most cultured mandibles without incision (n=8/10, ***Figure 10L***). Thus, cells in the AP region move the lingual-buccal axis to form the tongue frenum. Indeed, serial histological observation indicated that in some *Ofd1*$^{fl/WT}$*;Wnt1Cre*(HET) mice, the tongue frenum was absent at the most anterior part of mandible (***Figure 10N***), while the frenum was present at a slightly more posterior region in the same mouse (***Figure 10O, N***; were obtained from the same mutant mouse).

No activation of Hh or canonical Wnt signaling could be seen in the wild-type tongue frenum (***Figure 10—figure supplement 1I and J***), suggesting that Hh or canonical Wnt signaling is not involved in tongue frenum formation. In addition to *Lgr5/Tbx22*, it has been shown that the deletion of non-canonical Wnt signaling-related molecules, *Wnt5a* and *Ror2*, led to a lack of tongue frenum (***Liu et al., 2012***; ***Mossaad et al., 2018***). We found p-c-Jun (marker of non-canonical Wnt signaling) positive cells in wild-type tongue frenum at E12, and in AP region at E11.5 (***Figure 10P and Q***). Moreover, no tongue frenum formation was observed in wild-type mice, when the mandible was cultured with inhibitor of non-canonical Wnt signaling, U73122 (n=0/4; ***Figure 10S***). Thus, tongue frenum formation is under control of non-canonical Wnt signaling. The presence or absence of the tongue frenum was varied between *Ofd1*$^{fl/WT}$*;Wnt1Cre*(HET) mice, which is likely relied on X-inactivation. In fact, when we cultured mandibles of *Ofd1*$^{fl/WT}$*;Wnt1Cre*(HET)*;Hprt*$^{GFP}$ mice showing cells with inactivation of normal X-chromosome (Gfp-negative) at the AP region (***Figure 10T***, before culture), the tongue frenum was absent (***Figure 10U***, after culture). The Gfp-negative domain in *Ofd1*$^{fl/WT}$*;Wnt1Cre*(HET)*;Hprt*$^{GFP}$ mice were found to contain adipocytes. To understand whether the presence of adipose at AP region

affect tongue frenum formation, CNCC were replaced to adipose at AP region in wild-type mice, and cultured them. The tongue frenum was impaired in the cultured wild-type mandible with replacement (n=1/5; *Figure 10W and X*). Thus, tongue frenum formation is also reliant on which type of cells first reach to the AP region. Tongue frenum formation occurred when CNCC reach first, while it is inhibited when adipocytes reach first.

## Discussion

Tongue development contains multiple cellular processes including cell-cell interaction, cell migration, and differentiation. We found that contact between CNCC and mesoderm-derived cells are involved in differentiation of mesoderm-derived cells into myoblasts through Hh signaling in CNCC. The Hh signaling in CNCC is found to be under control of *Ofd1*. Failure of activation of Hh signaling in CNCC or contact between CNCC and mesoderm-derived cells results in mis-differentiation of mesoderm-derived cells into brown adipocytes. Our findings also suggest the possibility that after interacting with CNCC, mesoderm-derived cells also induce Hh signaling for their differentiation, which is likely evoked by CNCC with Hh signaling activity. Hh signaling in mesoderm-derived cells could not be activated by CNCC without Hh signaling. Without proper interaction between CNCC and mesoderm-derived cells, mesoderm-derived cells also fail to induce their Hh signaling. Thus, first cell-cell interaction between CNCC and mesoderm-derived cells is crucial for subsequent event in tongue development.

Our findings revealed that tongue morphology is determined by which type of cells first reach to the specific region, that affect subsequent tongue morphology. These tongue anomalies are observed in syndromic and non-syndromic condition in human (*Cobourne et al., 2019*; *Hill et al., 2021*; *Li et al., 2020*; *Yin and Zhao, 2020*). Similar tongue phenotypes were observed in wild-type mice with mechanical manipulations. These revealed that genetical or mechanical disturbances in these processes result in syndromic or non-syndromic tongue anomalies, respectively. Moreover, these cellular processes are under the control of X-inactivation.

The phenotypic variation seen in the tongues of *Ofd1*^fl/WT^;*Wnt1Cre*(HET) mice is likely caused by X-inactivation. However, *Ofd1*^fl/WT^;*Osr2Cre*(HET) mice display no apparent brown adipose tissue, even though X-inactivation is present in these mice. Instead, brown adipose tissue was observed in all *Ofd1*^fl^;*Osr2Cre*(HM) mice, in which no X-inactivation occurs. All cells in the *Osr2-Cre* expression domain were *Ofd1* mutant cells in *Ofd1*^fl^;*Osr2Cre*(HM) mice, whereas only a certain number of cells were *Ofd1* mutant cells in *Ofd1*^fl/WT^;*Osr2Cre*(HET) mice, due to X-inactivation. The *Osr2-Cre* domain only partially overlaps with CNCC in paryngeal arch region (see *Figure 2N''*). These results indicate that *Ofd1* deletion from all cells of the *Osr2-Cre* expression domain is necessary for obvious brown adipose tissue formation, while *Ofd1* deletion from some cells within the *Osr2-Cre* expression domain is not sufficient to induce morphological detectable brown adipose tissue formation. *Ofd1* mutant CNCC was observed as clusters within pharyngeal arch region (*Figure 4—figure supplement 1*). It is possible that abnormal differentiation of mesoderm-derived cells is occurred, when mesoderm-derived cells are interacted with clustered *Ofd1* (-) CNCC, but not single *Ofd1* (-) cell. Hence, clustered *Ofd1* (-) CNCC within the *Osr2Cre* expression domain through X-inactivation might not be enough amount to induce mis-differentiation of mesoderm-derived cells, while *Ofd1* deletion from all *Osr2-Cre*-expressing cells in *Ofd1*^fl^;*Osr2Cre*(HM) mice results in enough amount of clustered *Ofd1* (-) CNCC formation. In wild-type mice, mesoderm-derived cells could easily interact with *Ofd1* (+) CNCC, since *Ofd1* (+) CNCC spatter to entire pharyngeal region (*Figure 4A*). Further study is needed to elucidate the mechanisms of cluster formation of *Ofd1* (-) CNCC in *Ofd1* mutant mice. Furthermore, adipose tissue was often found to be surrounded by CNCC in *Ofd1*^fl/WT^;*Wnt1Cre*(HET) tongue (*Figure 2J*). It is possible that mesoderm-derived cells migrate to mandibular processes along with clustered *Ofd1* (-) CNCC.

It has been shown that CNCC act as a scaffolding structure for the organization of migrating myoblasts. Furthermore, the direction of tissue formation is determined after CNCC and myoprogenitors reach to the mandible, which might lead to a variety of tongue phenotypes in *Ofd1*^fl^;*Osr2Cre*(HM) mice.

Ciliopathies show prominent mixed symptoms in several organs, including the tongue. Our findings provide hints for possible future treatment in ciliopathies.

# Materials and methods

## Production and analysis of transgenic mice

All animal experiments were reviewed and approved by the Niigata University Institutional Animal Care and Use Committee (approval number SA00610, SD01308). *Ofd1*$^{fl/fl}$, *Smo*$^{fl/fl}$, *R26R*$^{LacZ}$, *R26R*$^{Yfp}$, *Wnt1-Cre*, *Osr2-Cre*, *Krt14-Cre*, *Sox10-Cre*, and *Ift88*$^{fl/fl}$ mice were produced as described by *Ferrante et al., 2006*, *Jeong et al., 2004*, *Soriano, 1999*, *Srinivas et al., 2001*, *Danielian et al., 1998*, *Lan et al., 2007*, *Yi et al., 2006*, *Matsuoka et al., 2005*, and *Haycraft et al., 2007*, respectively. *Hprt*$^{fl}$ mice (021427) were purchased from the Jackson laboratory. E0 was taken to be midnight prior to finding a vaginal plug.

## In situ hybridization

In situ hybridization was carried out to detect mRNAs using [$^{35}$S]UTP or DIG as described previously (*Ohazama et al., 2008*).

## Immunohistochemistry

Frozen sections were incubated with antibodies to GFP (ab13970; Abcam), p-c-Jun (#9261; Cell Signaling), KRT1-5 (14309-1-AP; Proteintech), Prdm16 (AF6295; R&D), UCP1 (U6382; Sigma), Myogen (ab1835, abcam), Agt (11992-1-AP; Proteintech), Trim14 (15742-1-AP; Proteintech), C/EBPβ (23431-1-AP, Proteintech), Pax3 (ab180754, abcam), Caspase-3 (9661S; Cell Signaling), Ki67 (12202; Cell Signaling), and MyoD (M351201; Dako). To detect anti-GFP or anti-MyoD antibodies, the sections were incubated with Alexa 488-conjugated secondary antibody. A tyramide signal amplification system (Perkin Elmer Life Science, Waltham, MA, USA) was used to detect the anti-KRT1-5 and anti-p-c-Jun antibodies. VECTASTAIN ABC Kit (Vector) was used to detect the anti-Ki67 antibodies.

## Quantitative PCR

E16.5 embryos were frozen and were sectioned into 12 μm thick slices. Then, sections were mounted on PEN membrane slides, which were stained by Toluidine blue. Sparse region was dissected with the laser microdissection system (Leica Microsystems) into a microcentrifuge tube cap placed directly beneath the section. The tube cap was filled with 75 μl of RNA later (Sigma-Aldrich). RNA was isolated using a RNeasy Mini Kit (QIAGEN). q-PCR was performed using GoTaq qPCR Master Mix (Promega) with the carboxy-X-rhodamine Dye and Rotor-Gen-Q (QIAGEN) detection system. All samples were run in triplicate for each experiment, and relative transcript abundance was normalized to the amount of GAPDH.

## Human sample

Approval for study on human subjects was obtained from the Niigata University (2018-0228). Informed consent was obtained to use tongue tissue. Clinical information including a three-generation pedigree was obtained. DNA was extracted using blood sample. Mutations in OFD1 were screened by using primer sequences covering all 23 exons. Two surgical specimens were selected from the surgical pathology files in the Division of Oral Pathology, Niigata University Graduate School of Medical and Dental Sciences.

## Organ culture

The mandible including the tongue region was separated from the head and body at E11.5. The mandible explants were immediately placed into a glass bottle containing culture media comprising 100% immediately centrifuged (IC) rat serum. IC rat serum was incubated at 56°C for 30 min to inactivate complement. The culture bottles were attached to a rotator drum and rotated at 14 rpm at 37°C while being continuously supplied with 5% $O_2$/95% $CO_2$ gas mixture. At the end of the culture period, explants were fixed with 4% PFA and processed for histological examination. U73122 (Selleck Chemicals) was used as 10 μM.

## Cell culture

Tissue around somite was dissected from mouse embryos at E9.5. Tissues were incubated in solution containing Dispase II (Roche) for 15 min at RT. Epithelium and mesenchyme were separated with

fine needles. The mesenchymal tissues were cut into pieces and plated in DMEM containing a high glucose (4.5 mg/ml) supplemented with 10% fetal bovine serum, 100 unit/ml penicillin, and 100 μg/ml streptomycin, and 50 μg/ml of ascorbic acid in a 5% $CO_2$ atmosphere at 37°C. With and without SAG (10 nM; Adipogen SA, Switzerland) or cyclopamine (10 μM, LC Laboratories, USA) and cultured for 10 days.

## Culture of human white adipocyte and myoblasts

Human skeletal muscle myoblasts (CC-2561) and subcutaneous preadipocyte cells (PT-5020) were purchased from Lonza (Lonza Walkersville). Human skeletal muscle myoblasts are isolated from the upper arm or leg muscle tissue of normal donors and are sold in second passage. Human subcutaneous preadipocytes are isolated from subcutaneous adipose tissue by enzymatic digestion and selective culturing techniques. Unpassaged cells are cryopreserved in a solution. All cells test negative for mycoplasma, bacteria, yeast, and fungi. HIV-1, hepatitis B, and hepatitis C are not detected for all donors and/or cell lots. Subcutaneous preadipocyte cells were induced to differentiate into terminal white adipocytes according to the manufacturer's protocol.

## Pseudo SEM

Pseudo SEM analysis was carried out as described previously (*Sandell et al., 2012*). Briefly, separated maxilla and mandible were fixed with 4% PFA, and then, stained with DAPI. Images were taken by confocal microscopy. The multiple sections were merged into a single projection view to generate the pseudo SEM image.

## Statistical analysis

Excel Toukei (ver. 6.0) was used for statistical analysis, which was done with a two-tailed unpaired Student's t test. $p < 0.05$ was considered statistically significant.

## Acknowledgements

This research was funded by the Japan Society for the Promotion of Science (JSPS; 17H01601, 20K18753, 20K18661, 21K10088).

## Additional information

### Funding

| Funder | Grant reference number | Author |
|---|---|---|
| Japan Society for the Promotion of Science | 17H01601 | Atsushi Ohazama |
| Japan Society for the Promotion of Science | 20K18753 | Akira Fujita |
| Japan Society for the Promotion of Science | 20K18661 | Takahiro Nagai |
| Japan Society for the Promotion of Science | 21K10088 | Yoko Ishida |

The funders had no role in study design, data collection and interpretation, or the decision to submit the work for publication.

### Author contributions

Maiko Kawasaki, Investigation, Writing – original draft; Katsushige Kawasaki, Finsa Tisna Sari, Takehisa Kudo, Jun Nihara, Madoka Kitamura, Takahiro Nagai, Vanessa Utama, Yoko Ishida, Fumiya Meguro, Alex Kesuma, Akira Fujita, Takayuki Nishimura, Yuan Kogure, Takeyasu Maeda, Sarah Ghafoor, Investigation; Satoshi Maruyama, Jun-ichi Tanuma, Pierre Corre, Brunella Franco, Resources; Yoshito Kakihara, Roman H Khonsari, Paul T Sharpe, Martyn Cobourne, Resources, Investigation; Atsushi Ohazama, Conceptualization, Resources, Data curation, Formal analysis, Supervision, Funding acquisition,

Investigation, Methodology, Writing – original draft, Project administration, Writing - review and editing

### Author ORCIDs
Maiko Kawasaki http://orcid.org/0009-0003-6273-5993
Takeyasu Maeda https://orcid.org/0000-0001-5317-7510
Paul T Sharpe https://orcid.org/0000-0003-2116-9561
Martyn Cobourne https://orcid.org/0000-0003-2857-0315
Atsushi Ohazama https://orcid.org/0000-0002-3496-0347

### Ethics
Approval for study on human subjects was obtained from the Niigata University (2018-0228) . Informed consent was obtained to use tongue tissue.

All animal experiments were reviewed and approved by the Niigata University Institutional Animal Care and Use Committee (approval number SA00610, SD01308).

### Decision letter and Author response
Decision letter https://doi.org/10.7554/eLife.85042.sa1
Author response https://doi.org/10.7554/eLife.85042.sa2

---

## Additional files

### Supplementary files
• MDAR checklist

### Data availability
All data generated or analysed during this study are included in the manuscript, supporting file and the uploaded document containing source numerical data for figures.

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
