## [Editor Report]

In this study, Kawasaki and colleagues made valuable finding that Hh signaling regulates mesoderm-derived cell fate determination through cell-cell interactions. They provided compelling evidence that disruption of primary cilia can lead to tongue development defects. They also showed solid evidence that correction of ciliary function may help to treat tongue malformations.

---

## [Decision Letter]

**Decision letter after peer review:**

Thank you for submitting your article "Coordinated multiple cellular processes in tongue development" for consideration by *eLife*. Your article has been reviewed by 3 peer reviewers, one of whom is a member of our Board of Reviewing Editors, and the evaluation has been overseen by Didier Stainier as the Senior Editor. The reviewers have opted to remain anonymous.

Essential revisions (for the authors):

1) This study is very descriptive in nature and will require extensive new experiments to address the molecular and cellular mechanism. The authors need to carefully develop a revision plan and outline additional studies to address reviewers' concerns.

2) Many of the conclusions are not supported by high-quality or definitive data. For example, there is no experimental evidence to support the conclusion that myogenic progenitors mis-differentiated into brown adipocytes in Ofd1 conditional knockout mice. This data must be provided to support the conclusion. What are the signals responsible for the cell fate switch from myogenic to adipogenic?

3) The title needs to be revised to reflect the mechanistic insights of the study,

4) The authors need to carefully review the use of terminology as indicated in the review.

*Reviewer #1 (Recommendations for the authors):*

However, this paper was mainly focused on the phenotype description of the Ofd cko mice and did not investigate deeply into the molecular changes that led to the CNCC and myogenic fate change. Moreover, some of the conclusions are not supported by high-quality data. Below are detailed comments:

1. On page 5 (lines 11 and lines 17-19), the conclusions "Ectopic bone and sparse tissue were observed in the tongue…" and "… immunohistochemistry analysis suggested that this tissue was brown adipose tissue, but not muscle or white adipose tissue (Figure 2A-C, S3)" need more supporting data, particularly for the immunohistochemistry analysis in Figure 1 and Figure S3. If the authors claim that there is ectopic bone (page 5, line 11), Figure 1 should provide markers for ectopic bone formation. Are Figure 1A, B, E, and F from comparable locations in control and mutant mice? In some sections, tooth germs are visible in the control, but not in the mutant. Do the mutants have an abnormal tooth phenotype as well? How can authors be sure that they are comparing the same locations? Furthermore, panels C, D, F, and G in Figure S3 are very high magnification images and therefore it is difficult to determine the original anatomical locations. The authors should include lower magnification images (e.g., Figures 2E and F) showing the region of the higher magnification images in Figure S3. On top of that, the Results section should describe the reasoning for using specific markers in the qPCR analysis in Figure 2A-C.

2. On page 6 (lines 7-9), the authors used Figure 2D to show "No YFP expression could be detected in the adipose tissue, indicating that ectopic brown adipose was derived from the mesoderm." Since Figure 2D only has YFP staining, it is not sufficient to support this conclusion. Co-localization of YFP signaling and brown adipose marker(s) is needed to support this conclusion.

3. The authors only showed YFP expression in Ofd1fl/WT;Wnt1Cre(HET);HprtGFP mice (Figure 3A) and did not show YFP expression in HprtGFP control mice. Similar expression patterns in the control samples are needed to draw the conclusion on page 8 that "YFP-expressing cells were observed in some region of the mandibular processes in Ofd1fl/WT;Wnt1Cre(HET);HprtGFP mice, indicating that normal CNCC was present in Ofd1fl/WT;Wnt1Cre(HET) mice (Figure 3A)". Moreover, control data is also needed to support the statement on page 9 that "It is likely that CNCC in the hypoglossal cord are randomly divided into two types of cell clusters (cells with and cells without Ofd1) due to the X-inactivation since X-inactivation is based on a random choice between the two X-chromosomes."

4. In Figure 3 D-E, it is difficult to see the relationship between MyoD+ and GFP+ cells because they are presented as two individual figures. The authors should provide co-localization of MyoD and GFP in the same sections to show their relationship more clearly.

5. The conclusion that "Thus, the interaction between CNCC and mesoderm-derived cells are reliant on Hh signalling in CNCC, which is essential for proper differentiation of mesoderm-derived cells." on page 10 needs higher quality data to support it. Similarly, Figure 3 I-J does not provide sufficient information on the relationship between GFP+ CNCCs and Ptch1+ cells as the expression patterns are shown separately as two individual images. Co-localization of GFP and Ptch1 on the same sections is needed to clarify whether reduced Hh signaling is occurring in the CNCCs or other cells. Again, Figure 3I-J is only showing the expression pattern of Ofd1fl/WT;Wnt1Cre(HET);HprtGFP samples. Control samples with the same staining should also be included to better evaluate the changes in CNCCs in Ofd cko mice.

6. The authors concluded that "ectopic brown adipose formation was caused by the lack of Hh signaling in CNCCs" on page 11. However, the authors did not further explore the detailed interaction between CNCCs and myogenic cells. Which CNCC-derived signaling molecules are affected in Ofd cko mice? Is it possible that these affected molecules from CNCCs lead to the fate change of myogenic cells to brown adipose?

7. The authors provided human OFD1 patient samples, which is a very valuable resource. However, although qPCR analysis of adipogenic genes has been performed, the presence of adipose tissue is not obvious from the HandE staining (Figure 6 C-F). Could the authors provide higher magnification of the histology sections to show the typical morphology of adipose cells in the patient samples? Or, if possible, could immunohistochemistry staining of adipose markers be performed on the patient sample sections? In addition, is the Hh signaling affected in the patient samples, too?

8. On page 15, the authors need more evidence for the conclusion that "After emergence of tongue swelling (E11.5), Osr2 expression was also found within the tongue swelling in wild-type mice (Figure 7Q, Lan et al., 2007). Osr2-expressing cells are likely to migrate into the tongue to form tongue connective tissue such as the lamina propria, tendon, and interstitial connective tissue since these tissues are formed by CNCC." The appearance of Osr2 can simply be the resident cells turning on Osr2 rather than the migration of Osr2+ cells, which the authors do not provide evidence for.

9. On page 15, the conclusion that "However, in Ofd1fl/WT;Wnt1Cre(HET) mice, these cells abnormally differentiate into osteoblasts due to the lack of Hh signaling, which migrate into the tongue when tongue formed." needs more supporting evidence. The functional requirement for Hh in osteogenic fate regulation needs to be experimentally tested. Could the authors test whether restoring Hh in Ofd1fl/WT;Wnt1Cre(HET) samples can rescue the CNCC defects? If transgenic mice are not available, primary cell cultures from Ofd1fl/WT;Wnt1Cre(HET) mice can also be used as an alternative.

*Reviewer #2 (Recommendations for the authors):*

1. The manuscript title and the overall conclusion that "tongue development requires coordinated multiple cellular processes" are uninformative. Formation of every organ in the mammalian body requires the coordination of multiple cellular and molecular processes. The major question is whether the data in this manuscript have uncovered significant new mechanisms coordinating the multiple cellular processes in tongue development. Please clearly summarize what new coordination mechanisms were uncovered.

2. The paper needs to use clear terms to distinguish tongue myoblast precursor cells from other mesoderm-derived cells in the mandible throughout the manuscript. For example, the statement "before mesoderm-derived cells enter the mandibular process they migrate as a single stream along the hypoglossal cord around E9-E10" (Lines 168-170) is wrong. Mesoderm-derived cells, including cranial mesoderm giving rise to endothelial cells and muscles other than tongue muscles, enter the mandibular process through different routes in different groups.

3. While Figure 2D shows that the sparse adipose tissue in the tongue was not derived from CNCCs, the manuscript only vaguely describes this tissue as derived from mesoderm but then implies, with no experimental evidence, that the adipose tissue is derived from aberrant differentiation of tongue myoblast precursor cells due to disruption of CNCC-mesoderm interactions in the hypoglossal cord. Please clarify the relationship between the adipose tissue and tongue myoblast precursor cells and provide clear experimental data supporting the conclusion.

4. Figure 2J shows that Myf5 expression was not present in the hypoglossal cord in the mutant embryo, but it is not known whether the tongue myoblast precursor cells failed to activate, or downregulated, Myf5 expression but still migrated through the hypoglossal cord or failed to migrate or died.

5. Data presented in Figure 2Q-S are not convincing to support the conclusion that disruption of CNCC-myoblast precursor interaction in the hypoglossal cord was the sole cause of ectopic brown adipose in the tongue because (1) the diagram in Figure 2Q suggests puncture of the mandibular arch tissues in addition to the hypoglossal cord and (2) the explant did not form a muscular tongue structure with proven brown adipose tissue derived from tongue myoblast precursor cells. In addition, although the Osr2-Cre lineage was shown to partly overlap with the hypoglossal cord at E10, Osr2 lineage cells are also in the tongue primordium at E11.5. Thus, the Osr2-Cre did not provide sufficient evidence for the hypoglossal cord interaction hypothesis (and the interpretation of the Smo/Osr2-Cre data in Line 245 also needs to take account of Osr2-Cre lineage cells in the tongue primordia in addition to the hypoglossal cord). Alternative methods, such as temporally controlled Ofd1 inactivation, might help provide the necessary evidence for the conclusion.

6. The manuscript describes the adipocyte differentiation as a result of "aberrant interaction" in Line 198 and "lack of proper cell-cell interaction between CNCC and mesoderm-derived cells" in Lines 201-202. Please clarify whether the adipocyte differentiation was due to loss or gain of a signal from the CNCCs and try to determine what that might be.

7. Lines 222-226, there is a likely alternative interpretation to the data in Figure 3: the colocalization of MyoD-negative domain with the GFP-negative domain in the hypoglossal cord could be due to Myf5/MyoD positive cells migrating around the Ofd1 mutant cells in the genetic mosaic situation rather than migrating normally but failing to express Myf5/MyoD. Furthermore, as myoblast precursor cells migrate through the hypoglossal cord, they encounter and interact with distinct CNCCs along the way rather than maintaining a one-on-one partnership. Thus, the same myoblast precursor could encounter wildtype allele-inactivated CNCCs and then Ofd1fl allele-inactivated CNCCs, and vice versa, along the way. Can the author clarify what mechanism could mediate the failure of MyoD expression in the myoblast precursor cells and if those same cells could reactivate MyoD expression and myoblast differentiation when they encounter Ofd1fl allele-inactivated CNCCs after they had contact with wildtype allele-inactivated Ofd1fl/+;Wnt1-Cre CNCCs in the hypoglossal cord? Identifying the mechanism involved would also help explain better why the Ofd1fl/WT;Osr2Cre mice do not have the ectopic adipose in tongue phenotype.

8. Line 263-265 and 277-278, please clarify the sentence "CNCC with Ofd1 mutation failed to migrate into this region (of site of future tongue swelling)". Does it mean that CNCCs contributing to the tongue swelling came from CNCCs outside of the mandibular arch together with the myoblasts? Multiple studies have shown that there were large numbers of CNCC death in the future tongue swelling region in CNCC-specific Smo mutant embryos. Thus, the lack of tongue swelling in the Ofd1(HM) embryos could well be due to CNCC cell death rather than a lack of CNCC migration to the area. Please check to see if CNCC survival in the mandibular arch of Ofd1(HM) mutant embryos from E9.5 to E11.5 was affected and explain why the authors interpret the data as a failure of CNCCs to migrate into the area. In addition, tongue swelling formation involves locally increased CNCC proliferation, which is also regulated by HH signaling. Thus, both apoptosis and defect in CNCC proliferation, rather than a defect in CNCC migration, might be the reason for the failure of tongue swelling formation in the HM mutant.

9. Lines 349-352, "unlike region for the emergence of the early tongue swelling, CNCC with Ofd1 mutation could migrate into the mandibular bone… some CNCC retained the capability of migration in Ofd1fl;Wnt1Cre(HM) mice" demonstrate complete lack of understanding of the mandibular developmental process. The tongue swelling and the mandibular bone form from CNCC-derived mandibular mesenchyme, rather than the CNCCs migrate into the swelling or the bone. Please rewrite many sentences throughout the text regarding CNCC "migration" in the tongue and bone development processes and only use the term "migration" for what it should mean when the cells move over other groups of cells from one place to another over some distance (such as myoblast precursors migrate through the CNCC-derived mesenchyme).

10. Lines 370-371, "In wild-type mice before emergence of tongue swelling (E11.0), Osr2-Cre activation was observed in the lingual region where mandible duplicated" does not make sense.

11. Figure 8I/I', it is unclear from the whole mount image which cells at what locations are labeled. Please provide images of serial sections of the mandible/tongue at the completion of the culture to show the locations of the labeled cells.

12. Lines 486-487, "Ofd1 mutation always occurred in the same region within the hypoglossal cord in Ofd1f/;Osr2Cre(HM) mice" is not supported by experimental evidence. In fact, the initial report on the Osr2Cre mice (Lan et al. 2007) described variability in Cre activity and pattern in those mice. Furthermore, because Osr2Cre activity only partly overlapped with the hypoglossal cord and tongue mesenchyme, it is not known why only some CNCCs at the hypoglossal cord/tongue mesenchyme exhibited Osr2Cre activity and whether the exact same cells in different Osr2-Cre mouse embryos exhibited Cre activity. Thus, the most logical interpretation of the phenotype of the Ofd1fl;Osr2Cre(HM) mice is that the differences resulted in variation in the pattern, level, and timing of Cre activity.

13. Sample numbers and developmental stages need to be included in figure legends.

*Reviewer #3 (Recommendations for the authors):*

The authors seem to misinterpret the results. In Figure 2D, the authors try to show that bone tissue in the tongue region is derived from the neural crest. The image is taken from the Ofd1fl/WT;Wnt1Cre(HET) and complete lack of GFP in the tongue means the cells in the tongue are all mesoderm-derived. Since it seems the tongue forms in the Ofd1fl/WT;Wnt1Cre(HET) to some extent, there is a possibility that the authors detected autofluorescence. It should be confirmed.

The objective of the experiment of making punctiform incisions is not clear. By making the incisions, it affects the tissue interaction but it induces bleeding or other tissue damage at the same time. It may not be ideal to include the results in the Figures, could be only used as supportive figures.

The word "hypoglossal cord" normally means a group of mesodermal cells that are derived from the occipital somites and moving into the branchial arches.

The interpretation of Figure 3F-H is not accurate since there is no confirmation the sections were obtained from the appropriate planes.

---

## [Author Response]

Essential revisions (for the authors):1) This study is very descriptive in nature and will require extensive new experiments to address the molecular and cellular mechanism. The authors need to carefully develop a revision plan and outline additional studies to address reviewers' concerns.2) Many of the conclusions are not supported by high-quality or definitive data. For example, there is no experimental evidence to support the conclusion that myogenic progenitors mis-differentiated into brown adipocytes in Ofd1 conditional knockout mice. This data must be provided to support the conclusion. What are the signals responsible for the cell fate switch from myogenic to adipogenic?

Many in vitro and in vivo data have been added as new data. We hope that these are enough for our conclusion. We also hope that mis-differentiation of mesodermal cells into adipocytes due to the lack of Hh signaling in CNCC is enough impact as a main conclusion. Our new findings raise the possibility that Hh signaling in mesodermal cells is also important for their differentiation as well as Hh signaling in CNCC, which has been added as a possibility in revised paper. We think that it is beyond the scope of this study to elucidate more detail of the signaling pathway to regulate mesodermal cell differentiation.

3) The title needs to be revised to reflect the mechanistic insights of the study,

The title has been changed.

4) The authors need to carefully review the use of terminology as indicated in the review.Reviewer #1 (Recommendations for the authors):However, this paper was mainly focused on the phenotype description of the Ofd cko mice and did not investigate deeply into the molecular changes that led to the CNCC and myogenic fate change. Moreover, some of the conclusions are not supported by high-quality data. Below are detailed comments:1. On page 5 (lines 11 and lines 17-19), the conclusions "Ectopic bone and sparse tissue were observed in the tongue…" and "… immunohistochemistry analysis suggested that this tissue was brown adipose tissue, but not muscle or white adipose tissue (Figure 2A-C, S3)" need more supporting data, particularly for the immunohistochemistry analysis in Figure 1 and Figure S3.

Immunohistochemistry data has been added as new Figure 2-S1 and 2-S2.

If the authors claim that there is ectopic bone (page 5, line 11), Figure 1 should provide markers for ectopic bone formation.

Expression of Runx2 (bone marker) have been added in new Figure 1-S2. However, bone part of results has been removed from revised paper, since ectopic bone is likely caused by abnormal mandibular formation, but not tongue formation. We would like to focus solely on tongue formation for this paper.

Are Figure 1A, B, E, and F from comparable locations in control and mutant mice? In some sections, tooth germs are visible in the control, but not in the mutant. Do the mutants have an abnormal tooth phenotype as well? How can authors be sure that they are comparing the same locations?

Figure 1A and 1E have been changed. Mutant mice show no obvious anomalies in molar tooth germs.

Furthermore, panels C, D, F, and G in Figure S3 are very high magnification images and therefore it is difficult to determine the original anatomical locations. The authors should include lower magnification images (e.g., Figures 2E and F) showing the region of the higher magnification images in Figure S3. On top of that, the Results section should describe the reasoning for using specific markers in the qPCR analysis in Figure 2A-C.

Lower magnification images have been added in new Figure 2-S1 and 2-S2.

The reasoning for using specific markers in the qPCR analysis have been added in Figure Legend of new Figure 2A-2C, 2-S1 and 2-S2 to keep the number of words within word limit.

2. On page 6 (lines 7-9), the authors used Figure 2D to show "No YFP expression could be detected in the adipose tissue, indicating that ectopic brown adipose was derived from the mesoderm." Since Figure 2D only has YFP staining, it is not sufficient to support this conclusion. Co-localization of YFP signaling and brown adipose marker(s) is needed to support this conclusion.

Double immunohistochemistry of Gfp and adipocyte marker has been added as new Figure 2J

3. The authors only showed YFP expression in Ofd1fl/WT;Wnt1Cre(HET);HprtGFP mice (Figure 3A) and did not show YFP expression in HprtGFP control mice. Similar expression patterns in the control samples are needed to draw the conclusion on page 8 that "YFP-expressing cells were observed in some region of the mandibular processes in Ofd1fl/WT;Wnt1Cre(HET);HprtGFP mice, indicating that normal CNCC was present in Ofd1fl/WT;Wnt1Cre(HET) mice (Figure 3A)". Moreover, control data is also needed to support the statement on page 9 that "It is likely that CNCC in the hypoglossal cord are randomly divided into two types of cell clusters (cells with and cells without Ofd1) due to the X-inactivation since X-inactivation is based on a random choice between the two X-chromosomes."

YFP immunohistochemistry in Hprt;Wnt1Cre (positive control) and Ofd1fl;Wnt1Cre(HM);HprtGFP (negative control) have been added in new Figure 4A and 4B. First sentence "YFP-expressing cells were observed in Ofd1fl/WT;Wnt1Cre(HET) mice (Figure 3A)" have been changed in new manuscript

4. In Figure 3 D-E, it is difficult to see the relationship between MyoD+ and GFP+ cells because they are presented as two individual figures. The authors should provide co-localization of MyoD and GFP in the same sections to show their relationship more clearly.

Gfp and MyoD double immunohistochemistry in Ofd1fl/WT;Wnt1Cre(HET);HprtGFP mice has been added in new Figure 4F.

5. The conclusion that "Thus, the interaction between CNCC and mesoderm-derived cells are reliant on Hh signalling in CNCC, which is essential for proper differentiation of mesoderm-derived cells." on page 10 needs higher quality data to support it.

Many in vitro and in vivo data have been added as new data. We hope that these are enough for our conclusion.

Similarly, Figure 3 I-J does not provide sufficient information on the relationship between GFP+ CNCCs and Ptch1+ cells as the expression patterns are shown separately as two individual images. Co-localization of GFP and Ptch1 on the same sections is needed to clarify whether reduced Hh signaling is occurring in the CNCCs or other cells. Again, Figure 3I-J is only showing the expression pattern of Ofd1fl/WT;Wnt1Cre(HET);HprtGFP samples. Control samples with the same staining should also be included to better evaluate the changes in CNCCs in Ofd cko mice.

Ptch1 and GFP double immunohistochemistry in Ofd1fl/WT;Wnt1Cre(HET);HprtGFP and Wnt1Cre;HprtGFP (control) has been added as new Figure 5D-5E.

6. The authors concluded that "ectopic brown adipose formation was caused by the lack of Hh signaling in CNCCs" on page 11. However, the authors did not further explore the detailed interaction between CNCCs and myogenic cells. Which CNCC-derived signaling molecules are affected in Ofd cko mice? Is it possible that these affected molecules from CNCCs lead to the fate change of myogenic cells to brown adipose?

Many in vitro and in vivo data have been added as new data. We hope that these are enough for our conclusion. It is extremely difficult to identify affected molecules from CNCCs for the fate change of myogenic cells after activation of Hh signaling in CNCC. Instead, our new findings raise the possibility that Hh signaling in mesodermal cells is also important for their differentiation as well as Hh signaling in CNCC, which has been added in revised paper. However, we think that it is beyond the scope of this study to deepen these.

7. The authors provided human OFD1 patient samples, which is a very valuable resource. However, although qPCR analysis of adipogenic genes has been performed, the presence of adipose tissue is not obvious from the HandE staining (Figure 6 C-F). Could the authors provide higher magnification of the histology sections to show the typical morphology of adipose cells in the patient samples? Or, if possible, could immunohistochemistry staining of adipose markers be performed on the patient sample sections? In addition, is the Hh signaling affected in the patient samples, too?

High magnification of HandE staining has been added as new Figure 9G and 9H. Immunohistochemistry data in human samples have been added as new Figure 9I-9L. Molecular analysis in Ofd1 mutant mice indicated that ectopic adipose in the tongue is similar to low thermogenic adipocyte. Histological feature of low thermogenic adipocyte is known to be different from that of typical brown adipose tissue. In fact, histological features of adipose in Ofd1 mutant mice is similar to that of low thermogenic, but not that of typical brown adipose. This has been mentioned in Results section.

Hh signal in CNCC is important, when CNCC and mesoderm-derived cells migrate within pharyngeal arch region before their differentiation. The tongue is not formed yet at the time. On the other hand, our human sample was obtained from tongue. It means that CNCC and mesoderm-derived cells already finished their migration and differentiation. We think that positive or negative of Hh signaling in CNCC at such later time should indicate other meanings.

8. On page 15, the authors need more evidence for the conclusion that "After emergence of tongue swelling (E11.5), Osr2 expression was also found within the tongue swelling in wild-type mice (Figure 7Q, Lan et al., 2007). Osr2-expressing cells are likely to migrate into the tongue to form tongue connective tissue such as the lamina propria, tendon, and interstitial connective tissue since these tissues are formed by CNCC." The appearance of Osr2 can simply be the resident cells turning on Osr2 rather than the migration of Osr2+ cells, which the authors do not provide evidence for.9. On page 15, the conclusion that "However, in Ofd1fl/WT;Wnt1Cre(HET) mice, these cells abnormally differentiate into osteoblasts due to the lack of Hh signaling, which migrate into the tongue when tongue formed." needs more supporting evidence. The functional requirement for Hh in osteogenic fate regulation needs to be experimentally tested. Could the authors test whether restoring Hh in Ofd1fl/WT;Wnt1Cre(HET) samples can rescue the CNCC defects? If transgenic mice are not available, primary cell cultures from Ofd1fl/WT;Wnt1Cre(HET) mice can also be used as an alternative.

Bone part of results has been removed from revised paper, since ectopic bone is likely caused by abnormal mandibular formation, but not tongue formation. We would like to focus solely on tongue formation for this paper.

Reviewer #2 (Recommendations for the authors):1. The manuscript title and the overall conclusion that "tongue development requires coordinated multiple cellular processes" are uninformative. Formation of every organ in the mammalian body requires the coordination of multiple cellular and molecular processes. The major question is whether the data in this manuscript have uncovered significant new mechanisms coordinating the multiple cellular processes in tongue development. Please clearly summarize what new coordination mechanisms were uncovered.

The title has been changed. The overall conclusion also has been changed.

2. The paper needs to use clear terms to distinguish tongue myoblast precursor cells from other mesoderm-derived cells in the mandible throughout the manuscript. For example, the statement "before mesoderm-derived cells enter the mandibular process they migrate as a single stream along the hypoglossal cord around E9-E10" (Lines 168-170) is wrong. Mesoderm-derived cells, including cranial mesoderm giving rise to endothelial cells and muscles other than tongue muscles, enter the mandibular process through different routes in different groups.

These have been changed in revised manuscript

3. While Figure 2D shows that the sparse adipose tissue in the tongue was not derived from CNCCs, the manuscript only vaguely describes this tissue as derived from mesoderm but then implies, with no experimental evidence, that the adipose tissue is derived from aberrant differentiation of tongue myoblast precursor cells due to disruption of CNCC-mesoderm interactions in the hypoglossal cord. Please clarify the relationship between the adipose tissue and tongue myoblast precursor cells and provide clear experimental data supporting the conclusion.

Many in vitro and in vivo data have been added as new data. We hope that these are enough for our conclusion.

4. Figure 2J shows that Myf5 expression was not present in the hypoglossal cord in the mutant embryo, but it is not known whether the tongue myoblast precursor cells failed to activate, or downregulated, Myf5 expression but still migrated through the hypoglossal cord or failed to migrate or died.

We agree with reviewer’s comment. However, it is difficult to determine these. Therefore, in new manuscript, these sentences have been changed.

5. Data presented in Figure 2Q-S are not convincing to support the conclusion that disruption of CNCC-myoblast precursor interaction in the hypoglossal cord was the sole cause of ectopic brown adipose in the tongue because (1) the diagram in Figure 2Q suggests puncture of the mandibular arch tissues in addition to the hypoglossal cord and (2) the explant did not form a muscular tongue structure with proven brown adipose tissue derived from tongue myoblast precursor cells. In addition, although the Osr2-Cre lineage was shown to partly overlap with the hypoglossal cord at E10, Osr2 lineage cells are also in the tongue primordium at E11.5. Thus, the Osr2-Cre did not provide sufficient evidence for the hypoglossal cord interaction hypothesis (and the interpretation of the Smo/Osr2-Cre data in Line 245 also needs to take account of Osr2-Cre lineage cells in the tongue primordia in addition to the hypoglossal cord). Alternative methods, such as temporally controlled Ofd1 inactivation, might help provide the necessary evidence for the conclusion.

The diagram has been changed, since old version was not correct.

Data showing the presence of muscle and adipose tissue confirmed by immunohistochemistry after giving incision have been added as new Figure 3P, 3Q. Many other in vitro and in vivo data have been added as new data. We hope that these are enough for our conclusion.

Unfortunately, we could not find any mice to create temporally controlled Ofd1 inactivation in CNCC during migration. However, only slight rescue of phenotypes in Ofd1 mutant using Sox10Cre indicate the importance of Ofd1 during migration. The data has been added in revised paper.

6. The manuscript describes the adipocyte differentiation as a result of "aberrant interaction" in Line 198 and "lack of proper cell-cell interaction between CNCC and mesoderm-derived cells" in Lines 201-202. Please clarify whether the adipocyte differentiation was due to loss or gain of a signal from the CNCCs and try to determine what that might be.

It is extremely difficult to identify signal from CNCCs for the differentiation of myogenic cells after activation of Hh signaling in CNCC. Instead, our new findings raise the possibility that Hh signaling in mesodermal cells is also important for their differentiation as well as Hh signaling in CNCC. These have been added in revised paper. However, we think that it is beyond the scope of this study to deepen these.

7. Lines 222-226, there is a likely alternative interpretation to the data in Figure 3: the colocalization of MyoD-negative domain with the GFP-negative domain in the hypoglossal cord could be due to Myf5/MyoD positive cells migrating around the Ofd1 mutant cells in the genetic mosaic situation rather than migrating normally but failing to express Myf5/MyoD. Furthermore, as myoblast precursor cells migrate through the hypoglossal cord, they encounter and interact with distinct CNCCs along the way rather than maintaining a one-on-one partnership. Thus, the same myoblast precursor could encounter wildtype allele-inactivated CNCCs and then Ofd1fl allele-inactivated CNCCs, and vice versa, along the way. Can the author clarify what mechanism could mediate the failure of MyoD expression in the myoblast precursor cells and if those same cells could reactivate MyoD expression and myoblast differentiation when they encounter Ofd1fl allele-inactivated CNCCs after they had contact with wildtype allele-inactivated Ofd1fl/+;Wnt1-Cre CNCCs in the hypoglossal cord? Identifying the mechanism involved would also help explain better why the Ofd1fl/WT;Osr2Cre mice do not have the ectopic adipose in tongue phenotype.

We agree with reviewer’s comment. However, it is difficult to prove these mechanisms. Therefore, these comments have been changed in new manuscript. On the other hand, our new data might suggest the possibility that being as a cluster of Ofd1(-) cells in Ofd1 cKO might be important to induce mis-differentiation of mesoderm rather than one-one contact. Furthermore, these data also raise the possibility that mesoderm-derived cells migrate into to mandibular processes along with CNCC cluster. Sentences describing these have been added.

8. Line 263-265 and 277-278, please clarify the sentence "CNCC with Ofd1 mutation failed to migrate into this region (of site of future tongue swelling)". Does it mean that CNCCs contributing to the tongue swelling came from CNCCs outside of the mandibular arch together with the myoblasts? Multiple studies have shown that there were large numbers of CNCC death in the future tongue swelling region in CNCC-specific Smo mutant embryos. Thus, the lack of tongue swelling in the Ofd1(HM) embryos could well be due to CNCC cell death rather than a lack of CNCC migration to the area. Please check to see if CNCC survival in the mandibular arch of Ofd1(HM) mutant embryos from E9.5 to E11.5 was affected and explain why the authors interpret the data as a failure of CNCCs to migrate into the area. In addition, tongue swelling formation involves locally increased CNCC proliferation, which is also regulated by HH signaling. Thus, both apoptosis and defect in CNCC proliferation, rather than a defect in CNCC migration, might be the reason for the failure of tongue swelling formation in the HM mutant.

We agree with reviewer’s comments. Sentences describing migration have been removed or changed. Apoptosis or cell proliferation were not altered in Ofd1 mutant mice, which has been added as new Figure 6-S2.

9. Lines 349-352, "unlike region for the emergence of the early tongue swelling, CNCC with Ofd1 mutation could migrate into the mandibular bone… some CNCC retained the capability of migration in Ofd1fl;Wnt1Cre(HM) mice" demonstrate complete lack of understanding of the mandibular developmental process. The tongue swelling and the mandibular bone form from CNCC-derived mandibular mesenchyme, rather than the CNCCs migrate into the swelling or the bone. Please rewrite many sentences throughout the text regarding CNCC "migration" in the tongue and bone development processes and only use the term "migration" for what it should mean when the cells move over other groups of cells from one place to another over some distance (such as myoblast precursors migrate through the CNCC-derived mesenchyme).

These sentences describing migration have been changed. Bone part of results was removed, since bone in tongue is likely caused by abnormal mandibular bone formation, but not tongue development. We would like to focus solely on tongue formation for this paper.

10. Lines 370-371, "In wild-type mice before emergence of tongue swelling (E11.0), Osr2-Cre activation was observed in the lingual region where mandible duplicated" does not make sense.

Bone part of results was removed, since bone in tongue is likely caused by abnormal mandibular bone formation, but not tongue development. We would like to focus solely on tongue formation for this paper.

11. Figure 8I/I', it is unclear from the whole mount image which cells at what locations are labeled. Please provide images of serial sections of the mandible/tongue at the completion of the culture to show the locations of the labeled cells.

Data showing section of explant with DiD injection has been added as Figure 10-S1F and -S1G. DiD labelled cells were located within Tbx22 and Lgr5 expression domain before culture and at tongue frenum region after culture

12. Lines 486-487, "Ofd1 mutation always occurred in the same region within the hypoglossal cord in Ofd1f/;Osr2Cre(HM) mice" is not supported by experimental evidence. In fact, the initial report on the Osr2Cre mice (Lan et al. 2007) described variability in Cre activity and pattern in those mice. Furthermore, because Osr2Cre activity only partly overlapped with the hypoglossal cord and tongue mesenchyme, it is not known why only some CNCCs at the hypoglossal cord/tongue mesenchyme exhibited Osr2Cre activity and whether the exact same cells in different Osr2-Cre mouse embryos exhibited Cre activity. Thus, the most logical interpretation of the phenotype of the Ofd1fl;Osr2Cre(HM) mice is that the differences resulted in variation in the pattern, level, and timing of Cre activity.

As Supplementary information 1 indicated, *Osr2Cre* activation (confirmed by LacZ staining in *Osr2Cre;R26R^LacZ^*) exhibited the three different patterns (tissue-specific, ectopic and ubiquitous pattern) (Lan et al., 2007). Approximately 30% of *Osr2Cre* displayed ectopic pattern or ubiquitous pattern. *Ofd1^fl^;Osr2Cre;R26R^YFP^* mice with ectopic pattern or ubiquitous pattern led to early lethal, as previous paper reported (Ferrante et al., 2006). Therefore, *Ofd1^fl^;Osr2Cre;R26R^YFP^* mice we can used in this study, only *Ofd1^fl^;Osr2Cre;R26R^YFP^* mice with tissue-specific pattern. Expression pattern and timing of Cre activity as tissue-specific pattern should be consistent in *Ofd1^fl^;Osr2Cre;R26R^YFP^* mice. These sentences have been added to clear this point in New Supplementary information 1.

*Osr2Cre* is unlikely overlapped with mesoderm, since *Osr2Cre;R26R^LacZ^* mice showed no LacZ positive cells in tongue muscle.

13. Sample numbers and developmental stages need to be included in figure legends.

These have been added into each Figure legends.

Reviewer #3 (Recommendations for the authors):The authors seem to misinterpret the results. In Figure 2D, the authors try to show that bone tissue in the tongue region is derived from the neural crest. The image is taken from the Ofd1fl/WT;Wnt1Cre(HET) and complete lack of GFP in the tongue means the cells in the tongue are all mesoderm-derived. Since it seems the tongue forms in the Ofd1fl/WT;Wnt1Cre(HET) to some extent, there is a possibility that the authors detected autofluorescence. It should be confirmed.

The picture has been replaced to new Figure 2J

The objective of the experiment of making punctiform incisions is not clear. By making the incisions, it affects the tissue interaction but it induces bleeding or other tissue damage at the same time. It may not be ideal to include the results in the Figures, could be only used as supportive figures.

Bleeding was not occurred at such early stage, since blood circulation is not well established yet at the stage. Immunohistochemistry data has been added to prove that adipose is formed in tissue with incision. It is hard to imagine that physical damage itself induces abnormal cell differentiation. So, we think that data is deserve to be as main Figure.

The word "hypoglossal cord" normally means a group of mesodermal cells that are derived from the occipital somites and moving into the branchial arches.

Sentence describing it have been changed.

The interpretation of Figure 3F-H is not accurate since there is no confirmation the sections were obtained from the appropriate planes.

Bright images (counter staining by hematoxylin) of in situ hybridization data have been added as new Figure 5A, 5B and 5C.